# Dense Metric Depth Estimation via Event-based Differential Focus Volume Prompting

**Boyu Li**[1,2]  **Peiqi Duan**[*1,2]  **Zhaojun Huang**[1,2]  **Xinyu Zhou**[3]  **Yifei Xia**[1,2]  **Boxin Shi**[*1,2]

[1]State Key Lab of Multimedia Info. Processing, School of Computer Science, Peking University
[2]National Eng. Research Center of Visual Tech., School of Computer Science, Peking University
[3]State Key Lab of General AI, School of Intelligence Science and Technology, Peking University
`{liboyu,duanqi0001,huangzhaojun,zhouxiny,yfxia,shiboxin}@pku.edu.cn`

## Abstract

Dense metric depth estimation has witnessed great developments in recent years. While single-image-based methods have demonstrated commendable performance in certain circumstances, they may encounter challenges regarding scale ambiguities and visual illusions in real world. Traditional depth-from-focus methods are constrained by low sampling rates during data acquisition. In this paper, we introduce a novel approach to enhance dense metric depth estimation by fusing events with image foundation models via a prompting approach. Specifically, we build Event-based Differential Focus Volumes (EDFV) using events triggered through focus sweeping, which are subsequently transformed into sparse metric depth maps. These maps are then utilized for prompting dense depth estimation via our proposed Event-based Depth Prompting Network. We further construct synthetic and real-captured datasets to facilitate the training and evaluation of both frame-based and event-based methods. Quantitative and qualitative results, including both in-domain and zero-shot experiments, demonstrate the superior performance of our method compared to existing approaches. Code and data will be available at `https://github.com/liboyu02/EDFV/`.

## 1 Introduction

Dense metric depth estimation is an important task to predict a dense map with absolute depth values of valid pixels. It plays a significant role in downstream applications such as augmented reality [25] and virtual reality [27]. Early single-image-based methods use hand-crafted features [56] or simple regression networks [12], but are limited by their robustness and generality. Recently, Image Foundation Models (IFM) trained on large datasets for depth estimation [3, 67] have been proposed. They are good at producing shapes and relative layouts of objects, but may suffer from scale ambiguities and visual illusions due to the lack of sufficient scene prior [33]. Although several studies [23, 48, 5] have incorporated camera intrinsic parameters to mitigate ambiguities for metric depth estimation, they may nonetheless encounter inaccuracies under certain conditions (Fig. 1 (c)).

For more accurate metric depth estimation, various cues have been utilized such as another view (stereo) [34], LiDAR measurements [57, 45, 33], and focus/defocus [40, 13]. Among them, stereo matching may have difficulties in handling spatial misalignments between frames [63]. LiDAR needs actively emitted laser pulses, and thus may pose problems to privacy and environment [44]. Depth from focus/defocus (DFF/DFD) methods rely on the fact that during focus sweeping, there is an optimal focusing timestamp for each point of the scene, where the Circle of Confusion (CoC) is the smallest (Fig. 1 (d)) [40]. They do not require auxiliary movements or equipment, thus ensuring

---

* Corresponding authors.

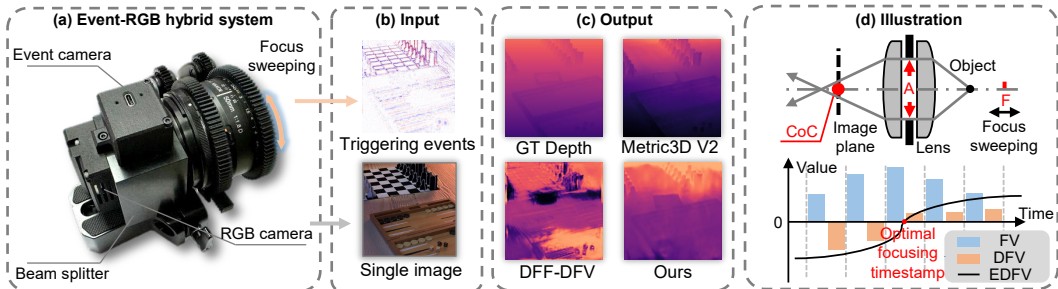

Figure 1: (a) We introduce a pipeline using both all-in-focus (AIF) images and focus sweeping triggered events for depth estimation. The input data can be captured with an event-RGB hybrid system. (b) Inputs of our method. (c) Comparison with other methods (single-image-based Metric3D V2 [23] and DFF method DFF-DFV [64]). (d) The illustration of the pinhole camera imaging model (top) and our proposed EDFV (bottom). Compared with frame-based FV and DFV, our method could better find the optimal focusing timestamp, resulting in more accurate depth estimation.

minimal spatial misalignment and applicable in most scenarios [64]. Most of them use Focus Volumes (FVs) and Differential Focus Volumes (DFVs) to assist depth estimation [59, 62]. However, at least 4 frames are usually required to achieve desirable results [40], increasing data acquisition difficulties, especially the consumption of operation time.

Event cameras are neuromorphic sensors inspired by human retina [31, 11]. The high temporal resolution enables them to record many temporal bins of the focus sweeping process in a short time. They have been employed in image restoration tasks such as deblurring [69], super-resolution [28], and all-in-focus (AIF) image reconstruction [38]. Recently, they have also been leveraged for sparse depth estimation [26]. These methods utilize event cameras to get better results with a shorter capture time. However, the sparsity of events makes it hard to get dense predictions. In contrast, DFF methods could extract dense information from images. These complementary characteristics motivate us to think: *Can we fuse the advantages of both images and events for dense metric depth estimation?*

There have been previous works achieving depth completion by prompting IFMs with sparse observations [45, 33]. However, utilizing event cameras for dense metric depth estimation still faces several challenges: (1) To better use the focus information encoded in events, a more efficient representation tailored for depth estimation is needed. (2) As events provide different sparse depth patterns from other sensors such as LiDAR because they mainly trigger at boundaries or textures, directly applying existing prompting strategies may fail to fully exploit their unique advantages and thus yield suboptimal performance. (3) No public datasets with ground truth (GT) depth, image focal stacks, and events are currently available.

In this paper, we propose a pipeline to combine events with images (Fig. 1 (a) & (b)) for more accurate dense metric depth estimation. We propose a new representation called **E**vent-based **D**ifferential **F**ocus **V**olume (EDFV) to guide sparse depth estimation. As shown in Fig. 1 (d), the high temporal resolution of events enables us to find the optimal focusing timestamp more accurately. Then we use EDFV to prompt the initial dense estimation results from IFM with our **E**vent-based **D**epth **P**rompting **N**etwork (EDPN) for the final prediction. To evaluate both frame-based and event-based DFF methods in unified reliable datasets, we synthesize two datasets in two different ways, and also capture datasets with real events. Above all, our contributions are as follows:

- We propose a novel representation EDFV constructed from events generated during focus sweeping for event-based sparse depth estimation, which accounts for the event triggering mechanism and enables more effective extraction of focus/defocus information.
- We design a prompting strategy to employ both EDFV and AIF images for dense metric depth estimation. The pipeline first predicts sparse depth maps from EDFV, and then leverages them to prompt dense depth estimation with IFMs through our prompting network.
- We construct two synthetic datasets including AIF images, focal stacks, events, and GT depth for the training and evaluation of both frame-based and event-based DFF methods. Besides, a semi-real dataset and a real dataset with challenging real-world scenarios are also captured for evaluation only. These datasets will be made public to facilitate future research.

We achieve at most 29.1% improvement in root mean square error (RMSE) on our proposed dataset compared with existing single-image-based and traditional DFF methods.

## 2 Related work

**Single image depth estimation.** Traditional methods [20, 56] use hand-crafted operators to estimate depth maps from single images. These methods usually fail to predict accurate depth in textureless areas. The learning-based approach learns deep depth cues from training data, thereby reducing the estimation error on similar test data [2, 70, 1]. However, they still struggle with zero-shot estimation in unseen scenes. With the rise of foundation models, some methods [3, 67] have used massive training data to achieve better single-image-based relative depth estimation results. To solve the scale ambiguity problem, more recent methods [68, 23, 48, 47, 16] take camera intrinsic parameters as additional inputs. But they still face challenges in real world due to limited scene prior, as shown in Fig. 1 (c). We seek to address the problem by prompting IFMs with information from focus/defocus.

**Focus measures and depth from focus/defocus.** The depth estimation methods based on focus/defocus take advantage of the lens effect that "the more the scene is away from the focus depth, the blurred in the image becomes" to realize the task of depth estimation from images [40]. Therefore, the focus measure operators they use for evaluating the "in-focus" degree of each pixel have a great influence on their performance [46]. Early researchers employ hand-crafted operators, such as gradient-based operators [15], wavelet-based operators [65], and discrete-cosine-transform-based operators [30] for the evaluation. Some methods estimate the depth map directly based on the blurred degree of local patches [43, 41], but the foreground/background ambiguity and the influence of the occlusion scene have an impact on the performance [55].

Deep learning methods are introduced to replace hand-crafted operators and improve performance [18, 40, 62]. DFV [64] is proposed to better utilize information by computing the first-order derivative on the stacked features. A more recent work [13] improves zero-shot performance by considering camera parameters in advance. Recently, HybridDepth [14] tries to incorporate both a pretrained traditional DFF method and a pretrained IFM as a relative depth estimator. However, these methods meet problems such as the dependence on focal stack numbers. We introduce events to record the focus sweeping process and take advantage of high speed to break through the above bottleneck.

**Event-guided depth estimation.** Event-based depth estimation mainly falls into three categories: monocular, stereo, and DFF. Early optimization-based monocular methods [29, 51] only output sparse or semi-dense results. Learning-based methods [19] have shown promising dense results. Although claimed as "monocular", these methods need camera or object motion during capture process, thus causing possible misalignment. Besides, the motion patterns may influence the quality of triggered events, thus affecting the estimation accuracy [35]. Stereo methods include event-based symmetric stereo [61, 42, 9] and event-intensity asymmetric stereo [71, 7]. Recently, some works [8, 37] employ foundation models from image domain to improve the generality. For event-based DFF methods, previous work [26] only predicts sparse depth from pure events. A more recent work [22] leverages existing grayscale video reconstruction approaches for depth estimation. Their method lacks the input image modality, which could constrain their zero-shot performance. In contrast, we aim to predict dense maps and improve zero-shot performance by fusing information from IFMs.

## 3 Preliminaries

In this section, we give a brief review of frame-based Focus Volume and Differential Focus Volume in Sec. 3.1, and formulate event-based Differential Focus Volume in Sec. 3.2.

### 3.1 Frame-based Focus Volume

According to the pinhole camera imaging model shown in Fig. 1 (d), objects away from the lens focus depth $F$ will exhibit defocus blur in the image [43]. The defocused image can be modeled as the result of the convolution operation between AIF image $\mathbf{I}$ and a spatial-variant Point Spread Function (PSF). The shape and blur degree of PSF are relevant to $\theta$, *i.e.*, radius of CoC, which is derived as:

$$\theta = A \cdot \frac{|\mathbf{D}(x,y) - F|}{\mathbf{D}(x,y)} \cdot \frac{f}{F-f}, \qquad (1)$$

where $A$ is the diameter of the lens aperture, $\mathbf{D}(x,y)$ represents the depth of the scene corresponding to pixel $(x,y)$, and $f$ is the focal length. According to Eq. (1), the closer $F$ to $\mathbf{D}(x,y)$, the smaller

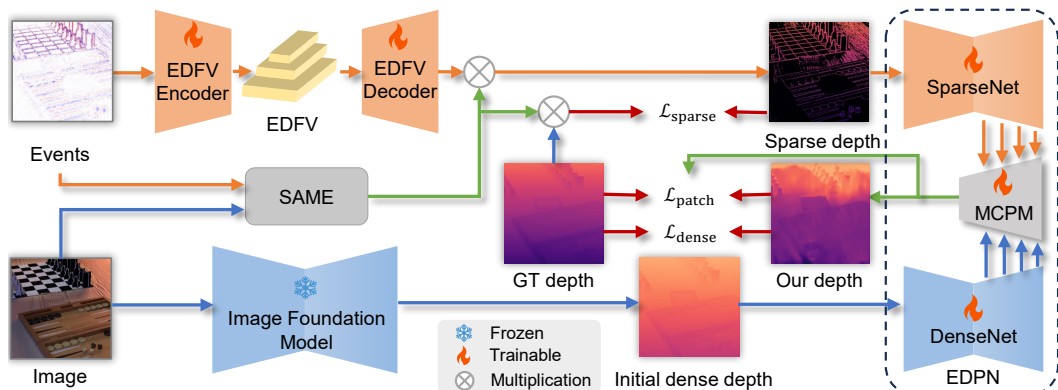

Figure 2: The pipeline of our method. Firstly, an image is fed into the IFM for an initial dense depth map. Concurrently, events are fed into an encoder to build multi-scale EDFVs. The SAME strategy is employed to extract spatial attention masks from images and events, which are multiplied with the output of the EDFV decoder to generate a sparse depth map. The two maps are subsequently fed into EDPN, where features are extracted from sparse and initial dense depth by SparseNet and DenseNet, respectively. Finally, the features are fused with MCPM to output our depth. $\mathcal{L}_{\text{sparse}}$ is calculated for depth consistency between sparse depth and masked GT (GT multiplied with the same spatial mask), $\mathcal{L}_{\text{dense}}$ between final depth and GT, as well as $\mathcal{L}_{\text{patch}}$ for multi-scale local consistency.

$\theta$ will be. When $F = \mathbf{D}(x, y)$, we have $\theta = 0$, which means the unique minimum point of $\theta$ corresponds to the depth of the object.

Frame-based DFF methods take images at $N$ different focal depths $F_i(i = 1, ...N)$, and find the optimal focused depth of each object in the scene. A common method is to use FV to store the "in-focus" degree of each pixel in each image. As discussed in Sec. 2, the focus measures can be divided into hand-crafted operators and deep-learning methods. Traditional methods find the maximum focus score among these FVs, while learning-based methods [62, 13] calculate the probability scores for the actual depth falling into each of these depths, and multiply them with these depths to get the result. Learning-based methods usually have better results. However, the feature magnitude of the sharpest pixel may not be salient in some circumstances due to noises and imperfect measurements [64]. Taking the single extremum into consideration, if taking the difference in the features, the polarity of the difference should be reversed around the best focal depth, and the sharpest pixel will be more significant to distinguish. This formulates the so-called DFV.

### 3.2 Event-based Differential Focus Volume

Event cameras are bio-inspired sensors that record intensity changes in the scene in an asynchronous manner. If the log-scale intensity of a pixel $\mathbf{I}(x, y, t)$ change exceeds a preset threshold $\epsilon$ at timestamp $t$, an event $e = (x, y, t, p)$ will be triggered, where $p \in \{+1, -1\}$ denotes the polarity indicating whether increasing or decreasing. This process can be formulated as:

$$p = \begin{cases} +1, & \log(\mathbf{I}(x, y, t)) - \log(\mathbf{I}(x, y, t - \Delta t)) > \epsilon, \\ -1, & \log(\mathbf{I}(x, y, t)) - \log(\mathbf{I}(x, y, t - \Delta t)) < -\epsilon. \end{cases} \tag{2}$$

**Observation 1:** Events triggered around the intensity-changing pixels at boundaries between objects of an image may experience a polarity reversal before and after focusing.

Here, the "intensity-changing pixels" mean pixels where themselves and the surrounding pixels do not have the same intensity. We assume a simple scene where there are only two objects, one in the foreground and one in the background, both with uniform intensity and depth. During the focus sweeping process, the intensity of a pixel $(x, y)$ on the background at any timestamp $t$ can be formulated as:

$$\mathbf{I}(x, y, t) = \mathbf{I}_b + \mathbf{I}_f \mathcal{P}_{\text{f}}(t), \tag{3}$$

where $\mathbf{I}_b$ and $\mathbf{I}_f$ are intensities of the background and foreground, respectively, and $\mathcal{P}_{\text{f}}$ is the value of the foreground PSF on pixel $(x, y)$. Then the log-scale intensity change is:

$$\Delta \log(\mathbf{I}(x, y, t)) = \log \frac{\mathbf{I}_b + \mathbf{I}_f \mathcal{P}_{\text{f}}(t)}{\mathbf{I}_b + \mathbf{I}_f \mathcal{P}_{\text{f}}(t - \Delta t)} = \log(1 + \mathbf{I}_f \frac{\mathcal{P}_{\text{f}}(t) - \mathcal{P}_{\text{f}}(t - \Delta t)}{\mathbf{I}_b + \mathbf{I}_f \mathcal{P}_{\text{f}}(t - \Delta t)}). \tag{4}$$

From the above equation, we derive:

$$\text{sign}(\Delta \log(\mathbf{I}(x, y, t))) = \text{sign}(\Delta \mathcal{P}_\text{f}(t)), \tag{5}$$

where $\text{sign}(\cdot)$ denotes the mathematical sign function.

**Observation 2:** The sign of the right side of Eq. (5) will change before and after focusing.

**Observation 3:** If the intensity-changing pixel is on the texture of a single object instead of an edge separating two objects, the triggered events around it may also experience a polarity reversal.

The derivation of these two observations can be found in the Appendix Sec. A. Based on the above observations, focus-sweeping triggered events inherently encode differential "in-focus" information. We exploit this characteristic to construct deep EDFV from events. Similar to DFV, EDFV may not perform well in textureless regions lacking salient intensity changes during sweeping. Moreover, the derivation shows that background events may reflect foreground intensity, differing from DFV. We will show how to address these challenges in our pipeline.

## 4 Method

The pipeline of our method is shown in Fig. 2. We first estimate initial dense depth map from a frozen IFM. Concurrently, we build EDFV with an encoder and predict a sparse metric depth map from events. The final estimation is generated by feeding two maps into our prompting network EDPN.

### 4.1 Initial dense depth estimation

Recent IFMs for depth estimation [66, 67, 3] take a single image as input, and output a dense depth map. Although maybe scale-ambiguous, they could preserve object boundaries and extract important semantic information of the scene. Therefore, we feed the image to IFM to estimate an initial dense depth map. In this work, we choose Depth Anything (DA) V2 [67] as our baseline model.

### 4.2 EDFV construction

In order to use the depth information encoded in events, a simple way could involve directly processing events with hand-crafted operators, such as the polarity changing timestamp for each pixel. However, this approach suffers from high noise sensitivity and may not output satisfactory results. Similar to deep frame-based FV and DFV discussed in Sec. 3.1 and the design of event-based neural representations [60], we first stack the events into $N$ temporal bins with their polarities, and then transform them into EDFV with a multi-scale convolutional encoder.

Previous studies [62] show the frame number of discrete focal stacks $N$ influences the performance of FV and DFV methods. But thanks to the high temporal resolution of events, when we sweep the focus ring, they can record continuous intensity changes for each pixel. Therefore, we can theoretically generate any number of event bins. Here we choose $N = 32$, and an ablation study of this number is provided in Sec. 5.5. Besides, we select the central depth of each bin as the candidate depths.

### 4.3 Sparse depth estimation

To solve the challenge that EDFV of a pixel may reflect depth values of surrounding pixels, we adopt a multi-scale decoder with residual convolution blocks to process the EDFV and efficiently estimate the sparse depth map. However, as the network itself could not identify effective pixels, noises of events in textureless areas may be scattered through these operations. To tackle this issue, we propose a Spatial Attention Mask Extraction (SAME) strategy to force the network to concentrate on the effective areas of events. Specifically, we use image gradients $\nabla \mathbf{I}$ and event density $\rho_e$ to build a spatial attention mask $\mathbf{M}$, which is multiplied with the output of EDFV decoder to get a sparse depth map. Besides, we add dilation operations to $\mathbf{M}$ to solve possible misalignments, which is:

$$\mathbf{M} = \text{Dilate}((\nabla \mathbf{I} > \epsilon_I) \cdot (\rho_e > \epsilon_e)), \mathbf{D}_s = \mathbf{M} \cdot (\sum_{i=1}^{N} \hat{\mathbf{S}}_i(\text{EDFV}) \cdot F_i), \tag{6}$$

where $\epsilon_I$ and $\epsilon_e$ are adjustable thresholds, $\hat{S}_i$ is the normalized probability score, and $\mathbf{D}_s$ is the sparse depth map. The choice of $\epsilon_I$ and $\epsilon_e$ would affect the effective regions of events, and thus affect

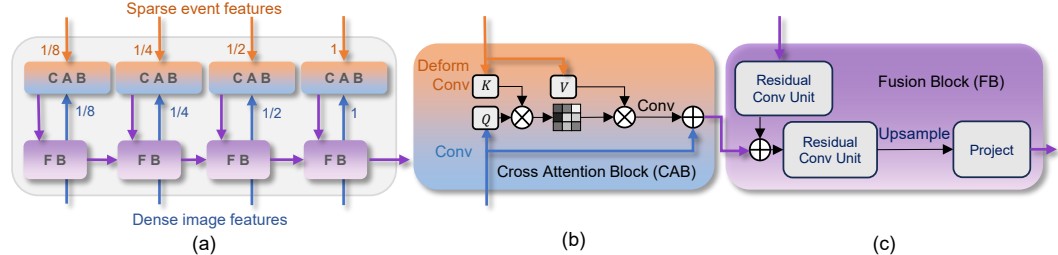

Figure 3: (a) The structure of MCPM, with CABs and FBs of 4 different scales. (b) The detailed structure of CAB. (c) The detailed structure of FB.

our final predictions. In order to improve the robustness of our network for the non-uniform and inconsistent event triggering thresholds and various noise levels in real world, we randomly set these parameters during our training, and select a fixed set of them for evaluation.

To provide a flatter distribution, we use softplus instead of softmax to normalize the scores, which is:

$$\hat{\mathbf{S}}_i = \frac{\ln(1 + \exp(\mathbf{S}_i))}{\sum_{j=1}^{N} \ln(1 + \exp(\mathbf{S}_j))}, \tag{7}$$

where $\mathbf{S}_i$ is the non-normalized score. Compared with softmax, this normalization improves depth estimation by interpolating among candidate depths [62].

### 4.4 Event-based depth prompting

Our prompting network leverages scale cues from event-based sparse depth to prompt the initial dense depth map for final results. Previous works [45, 33] mainly focus on LiDAR prompting approach, which differs in modality from event data sensitive to edges and textures. To bridge this gap, we employ two U-Net-based [53] networks, *i.e.*, SparseNet and DenseNet, to project sparse and initial dense depth maps into a unified feature embedding space. Considering their different characteristics, we add max-pooling layers to SparseNet to better densify the sparse information. In contrast, we add large kernel convolution layers to DenseNet for extracting more high-level features of the dense depth map. We collect features from decoders of different scales of SparseNet and DenseNet.

In order to solve the prompting challenges between two modalities, we further design a Multi-scale Cross-attention-guided Prompting Module (MCPM, as shown in Fig. 3). It consists of Cross Attention Blocks (CAB) and Fusion Blocks (FB) working at different scales. CAB first uses convolution layers to compute queries $\mathbf{Q}$ from dense image features $\mathbf{F}_I$, $\mathbf{K}$ and $\mathbf{V}$ from sparse event features $\mathbf{F}_e$. In order to handle the sparsity of events and possible misalignments between event and image features caused by motion as well as the challenge discussed in Sec. 3.2, we employ deformable convolutions [10] to generate $\mathbf{K}$ and $\mathbf{V}$. Then CAB use attention map as the matching score of $\mathbf{Q}$ and $\mathbf{K}$, and then use $\mathbf{V}$ to fill the output result. The above process can be formulated as:

$$\mathbf{Q} = \text{Conv}(\mathbf{F}_I), \mathbf{K}, \mathbf{V} = \text{DeformConv}(\mathbf{F}_e), \mathbf{D}_{\text{CAB}} = \text{Conv}(\text{Softmax}(\frac{\mathbf{Q}\mathbf{K}^\tau}{\lambda})\mathbf{V}) + \mathbf{F}_I, \tag{8}$$

where $\mathbf{D}_{\text{CAB}}$ is the output depth map of CAB, and $\lambda$ is a learnable scaling parameter.

Furthermore, we refine the output of CABs in a multi-scale way using FB. FB consists of Residual Convolution Units [32], and progressively upsample the representation by a factor of two in each fusion stage. The final output depth spatial resolution is the same as the input image.

### 4.5 Loss functions

We supervise our final results with a pixel-wise MSE loss between final output depth $\mathbf{D}$ and GT $\mathbf{D}_{\text{gt}}$:

$$\mathcal{L}_{dense} = \mathbb{E}[\|\mathbf{D} - \mathbf{D}_{\text{gt}}\|^2]. \tag{9}$$

In order to constrain our predicted sparse depth map from events to be accurate, we multiply the GT depth with $\mathbf{M}$, and calculate an MSE loss with sparse depth $\mathbf{D}_s$:

$$\mathcal{L}_{sparse} = \mathbb{E}[\|\mathbf{D}_s - \mathbf{D}_{\text{gt}} \cdot \mathbf{M}\|^2]. \tag{10}$$

Besides, according to previous studies [47], a patch-level edge-guided scale-shift invariant loss could help improve the local precision of models. Therefore, we adopt a multi-scale version formulated as:

$$\mathcal{L}_{patch} = \sum_{l=0}^{L} \beta_l \mathbb{E}_{\omega \in \Omega} \left[ \left\| \mathcal{N}_\omega(\mathbf{D}_{\text{CAB}}^{l,\omega}) - \mathcal{N}_\omega(\mathbf{D}_{\text{gt}}^{l,\omega}) \right\| \right], \tag{11}$$

where $L = 3$ corresponds to the level of MCPM, $\beta_l$ are weighting parameters, $\Omega$ is the set of selected image patches, $\mathcal{N}_\omega$ is the standardization operation over patch $\omega$, $\mathbf{D}_{\text{CAB}}^{l,\omega}$ are the output of CAB at level $l$ within $\omega$, and $\mathbf{D}_{\text{gt}}^{l,\omega}$ are the corresponding downsampled GT. $l = 0$ indicates original resolution.

The final loss function is the linear combination as:

$$\mathcal{L} = \mathcal{L}_{dense} + \alpha \mathcal{L}_{sparse} + \gamma \mathcal{L}_{patch}, \tag{12}$$

where $\alpha$ and $\gamma$ are parameters for balancing the contribution of different terms.

## 5 Experiments

### 5.1 Datasets

**Blender-Syn.** We follow DefocusNet [40] to use Blender [4] to render our first synthetic dataset. We put random objects in the scene, and wrap their surfaces with textures sampled from Poly Haven [49] to add their reality. For each scene, we shuffle and rescale the objects, and sweep the focus depth of the camera from 0.1 m to 30 m to generate a focus sweeping video of 500 frames, with an aperture f/1.4 and a focal length 100 mm. We select 150 scenes as training data and 60 scenes as test data, containing AIF images, GT depth maps, events and focal stacks. For each focal stack, we render 500 frames with different focal depths, and use them to simulate events using V2E [24]. Besides, to simulate possible motion in real data capturing, we add small position movements to each object.

**Sintel-Dr. Bokeh.** In order to overcome the limitation of object shapes and depth diversities in our first dataset, we use another large-scale single image depth estimation dataset, *i.e.*, Sintel [6], and generate focal stacks by adding synthetic blur to it. Our training set contains 130 scenes and test set contains 30 scenes. We use the bokeh rendering model Dr. Bokeh [58] to synthetically bokeh render these images at different focal depths. The blur strength parameter of Dr. Bokeh is set to 30 for more salient defocus cues. The biggest lens kernel size is set to 71, and the gamma parameter is 2.2. The event simulation process is the same as Blender-Syn.

**4DLFD-Semi-Real.** We further capture a semi-real dataset using 4D Light Field Dataset [21] as the source. It consists of 24 realistic scenes with densely sampled light field data and GT depth. After constructing focal stacks and AIF images from original data, we stack the frames into a video with 50 FPS to simulate a focus-sweeping process, and display it on the screen. Then we use a Prophesee EVK4 event camera to capture real events by shooting the screen.

**EDFV-Real.** We further capture a real-world test dataset with AIF images, focal stacks and events for frame-based and event-based methods. The hybrid camera system we use consists of a machine vision camera (HIKVISION MV-CA050-12UC) and an event camera (Prophesee EVK4). The two cameras are co-aligned with a beam splitter. Both cameras use one lens with a focal length of 16 mm (HIKVISION MVL-MF1628M-8MP). Since this lens did not exhibit noticeable FoV breathing during our observations, we did not account for this effect in data acquisition. Besides, we use checkboards to calculate the homography matrix to ensure alignment between these two cameras. We capture AIF images with aperture f/16, focal stacks with f/8, and events with f/2.4.

### 5.2 Experimental settings

**Metrics.** The evaluated metrics include RMSE (root mean square error), RMSE log (root mean square logarithmic error), AbsRel (absolute relative error: $|\hat{d} - d|/d$), log10 (absolute log10 error,

Table 1: Quantitative comparisons of in-domain metric depth estimation. $\uparrow (\downarrow)$ indicates the higher (lower), the better performance. The best performances are highlighted in **bold**, and the second best in underline. Left numbers denote results on Blender-Syn, and right numbers denote Sintel-Dr. Bokeh.

| Method | Type | Blender-Syn | | | | | | Sintel-Dr. Bokeh | | | | | |
|---|---|---|---|---|---|---|---|---|---|---|---|---|---|
| | | RMSE($\downarrow$) | AbsRel($\downarrow$) | log10($\downarrow$) | $\delta_1(\uparrow)$ | $\delta_2(\uparrow)$ | $\delta_3(\uparrow)$ | RMSE($\downarrow$) | AbsRel($\downarrow$) | log10($\downarrow$) | $\delta_1(\uparrow)$ | $\delta_2(\uparrow)$ | $\delta_3(\uparrow)$ |
| DefocusNet | DFF | 0.243 | 0.372 | 0.107 | 0.734 | 0.818 | 0.861 | 0.209 | 0.728 | 0.192 | 0.412 | 0.644 | 0.797 |
| DFF-FV | DFF | 0.184 | 0.223 | 0.062 | 0.862 | 0.907 | 0.926 | 0.160 | 0.661 | 0.109 | 0.766 | 0.863 | 0.898 |
| DFF-DFV | DFF | 0.186 | 0.250 | 0.062 | 0.871 | 0.906 | 0.923 | 0.134 | 0.569 | 0.104 | 0.738 | 0.861 | 0.907 |
| DDFS | DFF | 0.244 | 0.387 | 0.109 | 0.723 | 0.804 | 0.849 | 0.282 | 1.072 | 0.282 | 0.441 | 0.578 | 0.648 |
| HybridDepth | DFF | 0.089 | 0.123 | 0.051 | 0.823 | 0.925 | 0.969 | 0.273 | 0.657 | 0.295 | 0.233 | 0.393 | 0.540 |
| DA V2 | Mono | **0.063** | 0.089 | 0.035 | 0.865 | 0.956 | **0.989** | 0.297 | 0.482 | 0.361 | 0.330 | 0.419 | 0.472 |
| Metric3D V2 | Mono | 0.095 | 0.162 | 0.062 | 0.826 | 0.934 | 0.973 | 0.170 | 0.479 | 0.174 | 0.452 | 0.561 | 0.754 |
| DAC | Mono | 0.176 | 0.238 | 0.115 | 0.654 | 0.868 | 0.947 | 0.273 | 0.951 | 0.289 | 0.268 | 0.409 | 0.573 |
| Ours | DFF | 0.068 | **0.077** | **0.028** | **0.919** | **0.972** | 0.987 | **0.095** | **0.141** | **0.072** | **0.806** | **0.901** | **0.945** |

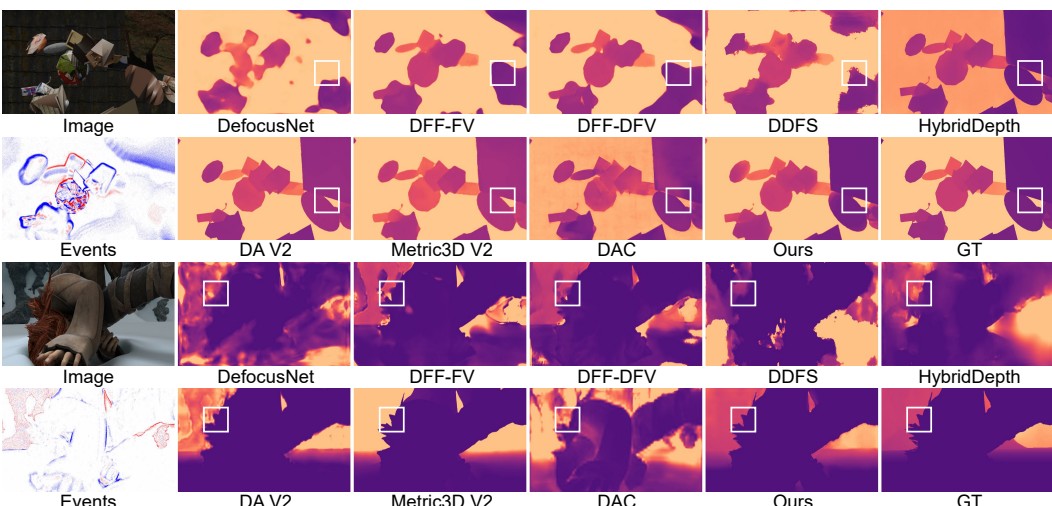

Figure 4: Qualitative in-domain experiment results on Blender-Syn (top 2 rows) and Sintel-Dr. Bokeh (bottom 2 rows) with inputs. We compare with DefocusNet [40], DFF-FV [64], DFF-DFV [64], DDFS [13], HybridDepth [14] and DA V2 [67], Metric3D V2 [23], and DAC [16].

$|\lg(\hat{d}) - \lg(d)|)$, and $\delta_i$ (percentage of $\max(\hat{d}/d, d/\hat{d}) < 1.25^i, i = 1, 2, 3$). For all the datasets, we select 5 frames with uniformly split focal depths as the input of frame-based DFF methods for training and evaluation, which is a common number used in these works.

**Training details.** We implement our method using the Pytorch framework and run on a single NVIDIA GeForce RTX 4090 GPU. We use AdamW [36] optimizer in the training phrase. For each dataset, we train for 700 epochs with initial learning rate $3 \times 10^{-4}$ and weight decay $1 \times 10^{-5}$. We randomly crop the input images into $320 \times 640$ resolution with random flipping afterwards. For loss functions, we set $\alpha = 0.1, \beta_0 = 1, \beta_1 = \beta_2 = \beta_3 = 0.1$ in all our experiments, $\gamma = 5$ for Blender-Syn dataset and $\gamma = 1$ for Sintel-Dr. Bokeh.

## 5.3 In-domain experiments

We report in-domain results on the first two datasets. For single-image-based methods, we compare with finetuned DA V2 from pretrained HyperSIM [52] ViT-L checkpoint, Metric3D V2 [23] from pretrained ViT-L checkpoint, and Depth Any Camera (DAC) [16] from pretrained outdoor ResNet101 checkpoint. For DFF methods, we evaluate DefocusNet [40], DFF-DFV [64], DDFS [13], and HybridDepth [14]. DFF-DFV's non-differentiable variant is denoted as DFF-FV. For fair comparison, all methods except DDFS are trained for the same epochs with their official code on our dataset. Since DDFS lacks public training code, we finetuned their released model using in-house code. Our table results use the DA checkpoint pretrained on relative depth with a ViT-L encoder.

The quantitative results are shown in Table 1, where we denote DFF methods as "DFF" and monocular methods as "Mono". Our method achieves the best performance in almost all metrics. It should be noted that DA V2 achieves commendable results on Blender-Syn, which may be attributed to

Table 2: Quantitative comparisons of zero-shot metric depth estimation.

| Method | Type | Blender-Syn | | | | | | Sintel-Dr. Bokeh | | | | | |
|---|---|---|---|---|---|---|---|---|---|---|---|---|---|
| | | RMSE(↓) | RMSE log(↓) | log10(↓) | $\delta_1$(↑) | $\delta_2$(↑) | $\delta_3$(↑) | RMSE(↓) | RMSE log(↓) | log10(↓) | $\delta_1$(↑) | $\delta_2$(↑) | $\delta_3$(↑) |
| DefocusNet | DFF | 0.425 | 0.783 | 0.292 | 0.135 | 0.391 | 0.608 | 0.518 | 1.504 | 0.585 | 0.123 | 0.204 | 0.275 |
| DFF-FV | DFF | 0.325 | 0.661 | 0.162 | 0.669 | 0.732 | 0.775 | 0.267 | 0.982 | 0.343 | 0.177 | 0.364 | 0.518 |
| DFF-DFV | DFF | 0.369 | 0.710 | 0.196 | 0.651 | 0.681 | 0.707 | 0.270 | 1.038 | 0.366 | 0.192 | 0.332 | 0.495 |
| DDFS | DFF | 0.495 | 1.120 | 0.377 | 0.287 | 0.361 | 0.448 | 0.706 | 1.852 | 0.726 | 0.203 | 0.231 | 0.251 |
| HybridDepth | DFF | 0.622 | 1.461 | 0.570 | 0.050 | 0.127 | 0.227 | 0.442 | 1.391 | 0.551 | 0.110 | 0.186 | 0.262 |
| DA V2 | Mono | 0.725 | 1.913 | 0.783 | 0.018 | 0.057 | 0.108 | 0.337 | 1.048 | 0.396 | 0.242 | 0.391 | 0.461 |
| Metric3D V2 | Mono | 0.294 | 0.535 | 0.207 | 0.343 | 0.625 | 0.777 | 0.322 | 1.174 | 0.469 | 0.262 | 0.369 | 0.466 |
| DAC | Mono | 0.652 | 1.488 | 0.594 | 0.075 | 0.142 | 0.198 | 0.515 | 1.474 | 0.581 | 0.144 | 0.228 | 0.285 |
| Ours | DFF | **0.148** | **0.282** | **0.081** | **0.697** | **0.878** | **0.944** | **0.233** | **0.685** | **0.253** | **0.333** | **0.466** | **0.560** |

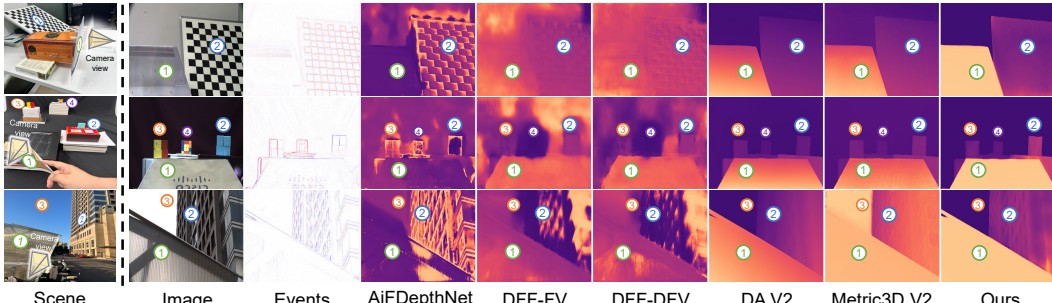

Scene | Image | Events | AiFDepthNet | DFF-FV | DFF-DFV | DA V2 | Metric3D V2 | Ours

Figure 5: Qualitative zero-shot experiment results on EDFV-Real dataset. We visualize inverse depth here. The first column is the scene overview, and the hand in the second row is just to indicate the thin paper. The numbers label the objects in each row. We compare with DFF methods AiFDepthNet [62], DFF-FV and DFF-DFV [64], and single-image-based DA V2 [67] and Metric3D V2 [23].

relatively simple shapes and scales. In contrast, Sintel-Dr. Bokeh contains more diverse scenes, so DFF methods may perform better. But our method still outperforms the best DFF method (DFF-DFV) in RMSE by over 29.1% on this dataset. Qualitative results are shown in Fig. 4. All the depth maps are clipped according to the depth range of focal stacks. It can be found that DFF methods are likely to produce artifacts due to low sampling rate and limited model capacity. Meanwhile, IFMs are good at predicting object boundaries but may fail to output accurate values due to scale ambiguity. Our prompting approach exhibits fewer artifacts and more accurate values.

## 5.4 Zero-shot experiments

We compare the zero-shot performance on Blender-Syn dataset with the model trained or finetuned on Sintel-Dr. Bokeh, and on Sintel-Dr. Bokeh with models trained on Blender-Syn. Quantitative results are shown in Table 2, which verify our model's superior zero-shot performance.

We further conduct zero-shot qualitative evaluations on 4DLFD-Semi-Real and EDFV-Real datasets. One example from 4DLFD-Semi-Real is shown in Fig. 1, with more in the Appendix Sec. D. Results on EDFV-Real are in Fig. 5. As other methods trained on our data perform poorly on these datasets, we use their official pretrained models. Our results use the Blender-Syn trained checkpoint. We designed our dataset to challenge single-image-based methods primarily in three ways: (1) Warped textures printed on thin paper to confuse singular and gradient depth. In the first and second rows, object ① exemplifies such cases, where single-image-based methods incorrectly predict gradient depths, while DFF-based and our method recover correct depth. Conversely, object ② in the first row is a real object for which single-image-based and our methods predict gradient depths. (2) Scale ambiguity via placement of similar-looking objects at varying depths. In the second row, objects ②, ③, and ④ differ in depth; only our method identifies them correctly. (3) Occlusion and overlap for misleading absolute metric scales. In the third row, objects ① and ② are distant objects (carport and building); traditional DFF methods fail due to complex textures. DA V2 estimates similar depths for both, Metric3D V2 errs on the sky region (object ③), while ours yields a plausible depth map.

## 5.5 Ablation studies

Results of ablation studies are in Table 3. All of the models are trained and tested on Blender-Syn.

**Prompting an IFM.** We show the effectiveness of prompting by the following experiments: removing IFM and directly feeding the image to DenseNet for feature extraction (denoted as "w/o IFM"); removing events or images from input (denoted as "w/o events" and "w/o image", respectively). The results show their performances drop as expected, among which "w/o image" has the most decline.

**Prompting network design.** The effectiveness of our proposed MCPM is proved by replacing it with simple add and convolution operations for fusion (denoted as "w/o MCPM"). Furthermore, we show the effectiveness of $\mathcal{L}_{\text{patch}}$ by removing it from our loss functions (denoted as "w/o $\mathcal{L}_{\text{patch}}$"). Besides, we conduct an ablation study where we replace the event inputs in our method with 32-frame focal stacks, denoted as "Ours-FS". Although this brings minor improvements to some metrics, it brings much more computational complexity to the whole pipeline (please refer to the Appendix Sec. C), which is not desirable for resource-limited devices.

**Temporal bin number.** We conduct ablation studies on the temporal bin number $N$. We denote our full model of bin number $i$ as "Full-$i$bin" ($i = 4, 8, 16$). As expected, the performance falls as $N$ decreases. Besides, the performance gain shrinks as the temporal bin number increases, which is consistent with observations in frame-based DFF methods. Notably, our "Full-4bin" model still defeats some DFF methods with 5 input frames in some metrics (refer to Table 1), demonstrating our method's advantage. We choose $N = 32$ in our final model.

**Alternative IFM checkpoints.** As our method supports flexible deployment with other IFM alternatives, we further evaluate the impact of IFM model checkpoint to our method. "Ours-DA-S" and "Ours-DA-B" denote our model with the DA V2 [67] checkpoint of ViT-Small encoder and ViT-Base encoder [50], respectively. In addition, "Ours-M3D-S", "Ours-M3D-L" and "Ours-M3D-G" denote our model with the Metric3D V2 [23] checkpoint of ViT-Small, ViT-Large and ViT-Giant backbones, respectively. Our final version is based on DA V2 of ViT-Large backbone. Although the performance based on smaller

Table 3: Quantitative results of ablation studies.

| Method | RMSE($\downarrow$) | AbsRel($\downarrow$) | $\delta_1(\uparrow)$ |
|---|---|---|---|
| w/o IFM | 0.122 | 0.177 | 0.828 |
| w/o events | 0.106 | 0.21 | 0.741 |
| w/o image | 0.216 | 0.381 | 0.579 |
| Ours-FS | **0.067** | 0.079 | **0.920** |
| w/o MCPM | 0.104 | 0.114 | 0.840 |
| w/o $\mathcal{L}_{\text{patch}}$ | 0.073 | 0.096 | 0.874 |
| Full-4bin | 0.162 | 0.446 | 0.705 |
| Full-8bin | 0.084 | 0.091 | 0.880 |
| Full-16bin | 0.071 | 0.083 | 0.909 |
| Ours-DA-S | 0.086 | 0.098 | 0.900 |
| Ours-DA-B | 0.076 | 0.086 | 0.905 |
| Ours-M3D-S | 0.122 | 0.143 | 0.813 |
| Ours-M3D-L | 0.077 | 0.096 | 0.872 |
| Ours-M3D-G | 0.085 | 0.103 | 0.855 |
| w/o pretrain | 0.126 | 0.223 | 0.864 |
| Ours | 0.068 | **0.077** | 0.919 |

checkpoints may degrade slightly, they still defeat most of the other methods (refer to Table 1). We also conduct an ablation study by excluding the influence of pretrained checkpoints, denoted as "w/o pretrain", where we train our method from scratch. The results show that even without any pretrained checkpoint, our method still outperforms most existing DFF methods in most metrics.

# 6 Conclusion

We propose a pipeline using focus sweeping triggered events and images to generate dense metric depth. Thanks to the high temporal resolution of events, we could build EDFV and get a sparse depth map. Then we could get a more accurate dense estimation via prompting IFMs through our prompting network. Experiments show great in-domain and zero-shot performance of our method.

**Limitations.** Although we have addressed potential minor misalignments and motion issues, we intend to explore alternative strategies, such as employing liquid lenses [39] to adjust focus depth in future research. Besides, while our method demonstrates excellent zero-shot performance with current training datasets, scaling training data akin to IFMs could improve outcomes. Additionally, in low-light or highly-smooth scenarios, our method may suffer from low-quality events providing insufficient cues. Last but not least, when applying to high-speed scenarios, the synchronization of sensors could be an important issue as we need the frame and events to capture the scene with similar timing, and a specifically designed electronic system [72] could be the solution. We aim to work on enhancing the pipeline and generalizing our method further.

## Acknowledgments

This work was supported by National Natural Science Foundation of China (Grant No. 62136001, 62088102, 62402014), Beijing Natural Science Foundation (Grant No. L233024), and Beijing Municipal Science & Technology Commission, Administrative Commission of Zhongguancun Science Park (Grant No. Z241100003524012). Peiqi Duan was also supported by China National Postdoctoral Program for Innovative Talents (Grant No. BX20230010) and China Postdoctoral Science Foundation (Grant No. 2023M740076). The authors thank `https://openbayes.com/` for providing computing resources.

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

## Appendix

Our Appendix is organized as follows: First of all, we give additional mathematical derivation in Sec. A. Dataset construction details are provided in Sec. B. Computational efficiency comparison is discussed in C. Finally, we show more qualitative and quantitative results on our four datasets in Sec. D.

## A  Additional mathematical derivation

**Observation 2:** The sign of the right side of Eq. (5) in the main paper will change before and after focusing.

Suppose the pixel $(x_0, y_0)$ is in the foreground, and let $r = \sqrt{(x - x_0)^2 + (y - y_0)^2}$ be the distance of $(x_0, y_0)$ to $(x, y)$. When the radius of CoC, $\theta < r$, the pixel $(x, y)$ will not be affected by bokeh effects, so $\mathcal{P}_f(x, y, t) = 0$.

When $\theta \geq r$, according to previous studies [54, 17], PSF function of $(x_0, y_0)$ can be approximated as a Gaussian function, which can be formulated as:

$$\mathcal{P}_f(x, y, t) = \frac{T}{2\pi\sigma^2} \exp\left(-\frac{(x - x_0)^2 + (y - y_0)^2}{2\sigma^2}\right) > 0, \tag{13}$$

where $T$ is the amplitude, and $\sigma$ is the spread parameter proportional to $\theta$, *i.e.*, $\sigma = k\theta$.

Combining Eq. (1) in the main paper and the above analysis, we have: When $F$ approaches from a small distance to the foreground depth $\mathbf{D}_f$, $\theta$ will gradually transfer from a large value to $r$, and $\mathcal{P}_f(x, y, t)$ will decrease from a positive value to 0, *i.e.*, $\Delta(\mathcal{P}_f(x, y, t)) < 0$. Similarly, when $F$ moves from $\mathbf{D}_f$ to a larger value, $\theta$ will increase, and thus $\Delta(\mathcal{P}_f(x, y, t)) > 0$.

**Observation 3:** If the intensity-changing pixel is on the texture of a single object instead of an edge separating two objects, the triggered events around it will also experience a polarity reversal before and after focusing.

Suppose the texture splits the object into two parts with individual uniform intensity $\mathbf{I}_1$ and $\mathbf{I}_2$, respectively. Then we derive the intensity of a pixel $(x_1, y_1)$ in Part 1 can be formulated as:

$$\mathbf{I}(x_1, y_1, t) = \mathbf{I}_1 + \mathbf{I}_2 \mathcal{P}(t), \tag{14}$$

where $\mathcal{P}(t)$ is the PSF of the object. Then the log-scale intensity change can be derived as:

$$\Delta \log(\mathbf{I}(x_1, y_1, t)) = \log \frac{\mathbf{I}_1 + \mathbf{I}_2 \mathcal{P}(t)}{\mathbf{I}_1 + \mathbf{I}_2 \mathcal{P}(t - \Delta t)} = \log(1 + \mathbf{I}_2 \frac{\mathcal{P}(t) - \mathcal{P}(t - \Delta t)}{\mathbf{I}_1 + \mathbf{I}_2 \mathcal{P}(t - \Delta t)}). \tag{15}$$

Therefore,

$$\operatorname{sign}(\Delta \log(\mathbf{I}(x_1, y_1, t))) = \operatorname{sign}(\Delta \mathcal{P}(t)). \tag{16}$$

Similar to the above derivation, for a pixel $(x_2, y_2)$ in Part 2, we have

$$\operatorname{sign}(\Delta \log(\mathbf{I}(x_2, y_2, t))) = \operatorname{sign}(\Delta \mathcal{P}(t)). \tag{17}$$

So the intensity changes in both parts will undergo a polarity reversal before and after focusing. If these changes trigger events, a polarity reversal will also occur.

## B  More dataset construction details

**Blender-Syn.** We first select geometric objects, and then rescale and rotate them randomly in the scene. Then, the objects are put at different locations within depth range from 1m to 30m. After that, we wrap them with textures sampled from Poly Haven [49]. Finally, we sweep the focus depth of our camera from 0.1m to 30m to generate a focus sweeping video of 500 frames, with an aperture f/1.4 and a focal length 100mm. For each frame, we add slight motion and rotation to each object in the scene to simulate the shaking in real world. Besides, the AIF image is rendered using a small aperture (f/1000). The scenes are rendered using Blender [4] Cycles engine with reflection turned on, but with shadows off. Each image has a resolution $720 \times 1280$. Although the scene seems unrealistic

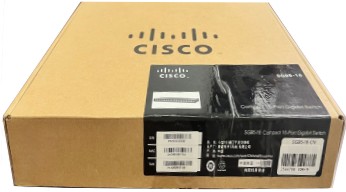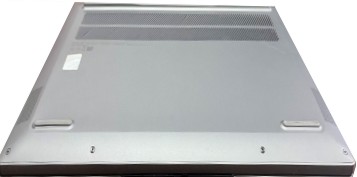

Figure 6: Example patterns we use for capturing EDFV-Real dataset.

Table 4: V2E parameters we adjust for simulating realistic events.

| Parameter | Description | Our value | Default value |
|---|---|---|---|
| shot-noise-rate-hz | Temporal noise rate of events in darkest parts of scene. | 0.1 | 0.001 |
| leak-rate-hz | Leak event rate per pixel in Hz. | 0.5 | 0.01 |

to some extent, our main purpose is to guide the models to learn focus/defocus cues instead of spatial and shape cues.

In order to improve the input information integrity, we evenly split the rendered frame focal stacks into 5 consecutive segments, and choose one frame from each segment as the input to frame-based DFF methods. Besides, for our method, we use V2E [24] to simulate events from the focal stack video. In order to add noises with reality, we manually adjust some parameters of V2E as shown in Table 4. The AIF image is used to be fed into single-image-based methods.

**Sintel-Dr. Bokeh.**    Since the GT of test set for Sintel [6] dataset has not been released, we randomly sample its training set as our dataset. Each image has a resolution $436 \times 1024$. We use their AIF images and GT depths as input to the bokeh rendering method Dr. Bokeh [58]. The blur strength parameter of Dr. Bokeh is set to 30 for more salient defocus cues. The biggest lens kernel size is 71, and the gamma parameter is 2.2. The focus depth is swept from 3.1m to 30m. After generating the focus sweeping video, the post process for generating input to methods is the same as Blender-Syn.

**4DLFD-Semi-Real.**    We follow AiFDepthNet [62] to generate AIF images and focal stacks from the light field data of 4D Light Field Dataset [21]. Then we concatenate the focal stacks into a video with 50 FPS, and play the video on a screen to simulate the focus sweeping process. A Prophesee EVK4 event camera is employed to record the process and generate real events for this semi-real dataset, which is for zero-shot testing only. It should be noted that the original data resolution is $512 \times 512$, while our captured events have a resolution $742 \times 720$. When evaluating our method on this dataset, we first interpolate the input images to the event resolution, and finally resize the output depth to the original resolution for computing metrics.

**EDFV-Real.**    In order to evaluate the performance of single-image-based, DFF and our methods in real-world scenarios, we further capture a real dataset with our hybrid camera system. Our hybrid camera system consists of a machine vision camera (HIKVISION MV-CA050-12UC) and an event camera (Prophesee EVK4). The two cameras are co-aligned with a beam splitter. Both of them use a 16 mm lens (HIKVISION MVL-MF1628M-8MP). Since this lens did not exhibit noticeable FoV breathing during our observations, we did not account for this effect in data acquisition. Besides, we use checkboards to calculate the homography matrix to ensure alignments between these two cameras before capturing.

During capture, we manually set scenes to "cheat" single-image-based methods. One way is to print warped textures on a thin paper to confuse them with singular and gradient depths. Some patterns we use are shown in Fig. 6. These patterns successfully mislead single-image-based methods (refer to qualitative results), while they can be distinguished by our method.

# C Computational efficiency

We compare the number of parameters, Multiply-Accumulate Operations (MACs) and inference time with other methods in Table 5. The MACs and inference times are calculated with a unified resolution $224 \times 224$, and all DFF methods use 5 frames as input. The runtimes are reported from a signle NVIDIA RTX 4090 GPU. Note the statistics of HybridDepth and our method in the table has excluded IFM. From the comparison, we can observe that our method introduces only $5.96 \div 332.68 \approx 1.79\%$ additional parameters and $15.44 \div 109.78 \approx 14.06\%$ additional MACs compared with the original DA V2. Besides, the runtime of our method is comparable to most other methods.

Additionally, we add the statistics of the ablation study of "Ours-FS" in Sec. 5.5, where we replace the input event with 32-frame focal stacks. It can be observed that this could bring $20.41 \div 5.96 \approx 3.42$ times parameter numbers and $134.91 \div 15.44 \approx 8.74$ times MACs to the whole pipeline, which is not desirable for resource-limited devices. Instead, our event-based method can achieve a balance between accuracy and efficiency, and is more practical to deploy on resource-limited devices.

Table 5: Computational efficiency comparison with other methods.

| Method | Type | Params(M) | MACs(G) | Time(ms) |
|--------|------|-----------|---------|----------|
| DefocusNet | DFF | 3.72 | 29.57 | 13.69 |
| DFF-FV | DFF | 18.40 | 20.54 | 27.35 |
| DFF-DFV | DFF | 18.40 | 20.54 | 28.40 |
| DDFS | DFF | 25.44 | 72.39 | 3.73 |
| HybridDepth | DFF | 34.97 | 24.11 | 32.96 |
| DA V2 | Mono | 332.68 | 109.78 | 23.59 |
| Metric3D V2 | Mono | 302.92 | 72.03 | 55.14 |
| DAC | Mono | 47.75 | 14.16 | 42.37 |
| Ours-FS | DFF | 20.41 | 134.91 | 28.40 |
| Ours | DFF | 5.96 | 15.44 | 17.40 |

# D More results

More qualitative results with reconstructed point clouds of in-domain experiments on Blender-Syn dataset are in Fig. 7 and Fig. 8, and Sintel-Dr. Bokeh dataset are in Fig. 9. Qualitative results of zero-shot experiments on Blender-Syn are shown in Fig. 10, and Sintel-Dr. Bokeh are shown in Fig. 11. From the comparison, we can observe the more accurate prediction results of our method.

Zero-shot experiment quantitative results on 4DLFD-Semi-Real dataset are shown in Table 6, and qualitative results in Fig. 12. Note that although our method may predict less accurate results in closer regions compared with Metric3D V2, possibly due to fewer textures, we achieve more accurate predictions in farther regions.

Figure 13 shows more qualitative results on EDFV-Real dataset. In the first scne, our method successfully distinguishes the pattern printed on the paper (object ①), while DA V2 and Metric3D V2 predict a gradient depth. In the second scene, we place three ducks of different sizes (objects ①, ② and ③) at a unified depth from the camera view. DA V2 predicts different depths for them due to scale ambiguities, while our method yields a reasonable result.

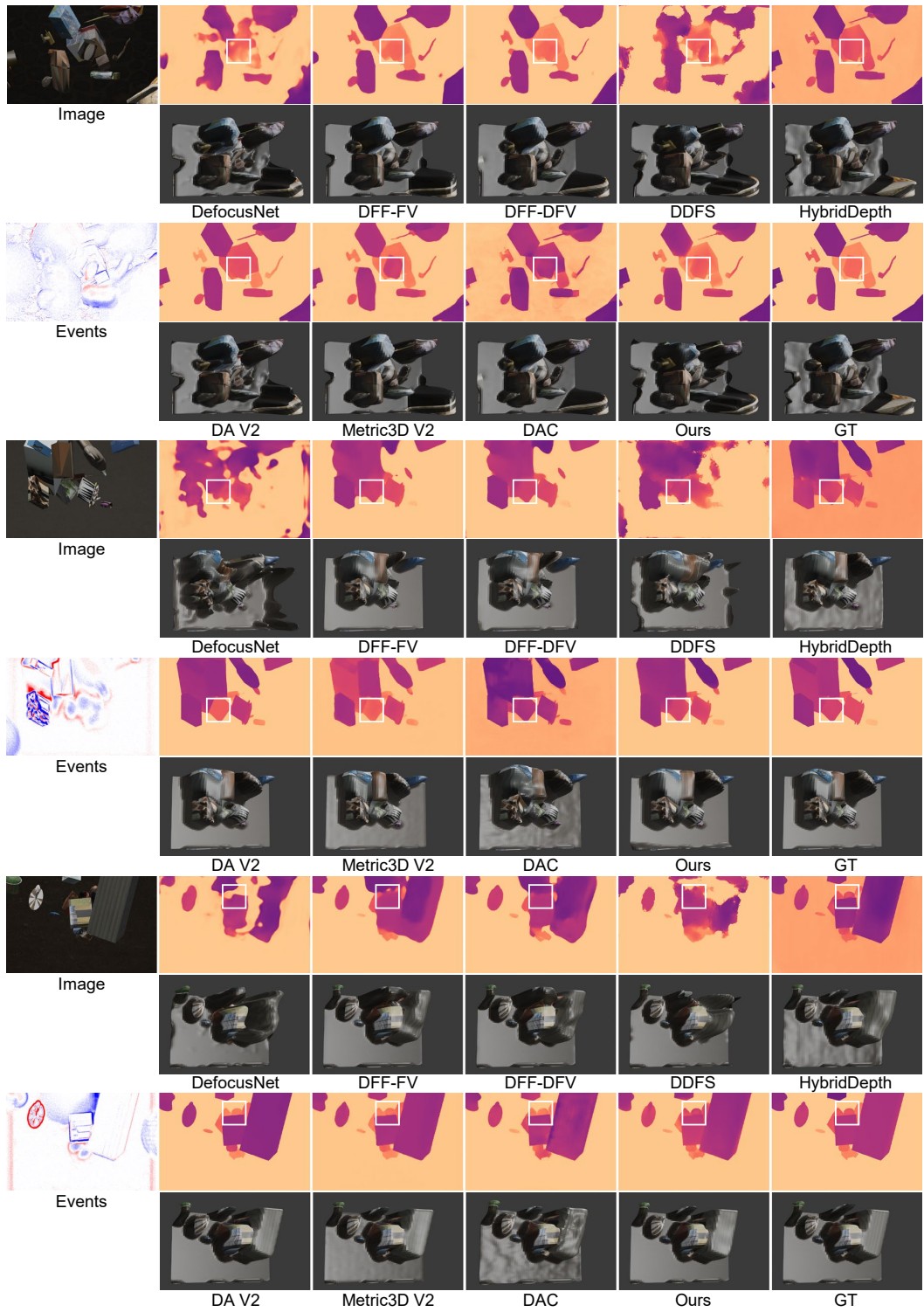

Figure 7: More qualitative results of in-domain experiments on Blender-Syn dataset with the input image and events (Part I). Each consecutive 4 rows corresponds to one test sample. Each odd row shows the input RGB image/events and predicted depth maps. Each even row shows reconstructed point clouds. We compare with DefocusNet [40], DFF-FV [64], DFF-DFV [64], DDFS [13], HybridDepth [14], DA V2 [67], Metric3D V2 [23], and DAC [16]. Please zoom in for more details.

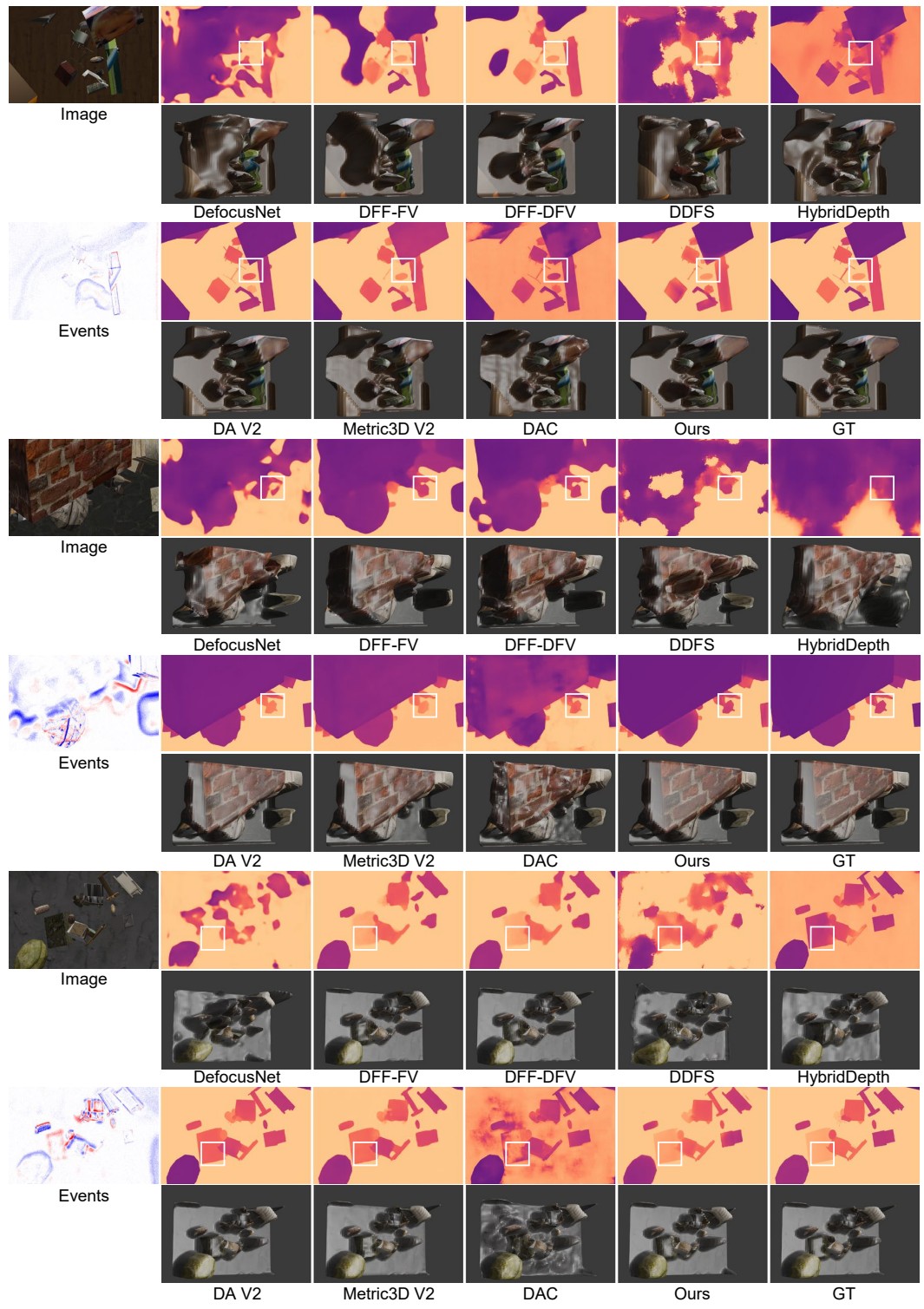

Figure 8: More qualitative results of in-domain experiments on Blender-Syn dataset with the input image and events (Part II).

Table 6: Quantitative comparisons of zero-shot metric depth estimation on 4DLFD-Semi-Real.

| Method | Type | RMSE(↓) | RMSE log(↓) | log10(↓) | $\delta_1(\uparrow)$ | $\delta_2(\uparrow)$ | $\delta_3(\uparrow)$ |
|---|---|---|---|---|---|---|---|
| DFF-FV | DFF | 1.979 | 0.198 | 0.070 | 0.680 | 0.888 | 0.949 |
| DFF-DFV | DFF | 1.943 | 0.186 | 0.064 | 0.711 | 0.902 | 0.953 |
| DDFS | DFF | 1.680 | 0.167 | 0.060 | 0.772 | 0.918 | 0.956 |
| HybridDepth | DFF | 1.573 | 0.135 | 0.048 | 0.818 | 0.940 | 0.957 |
| DA V2 | Mono | 2.997 | 0.227 | 0.088 | 0.561 | 0.897 | **0.958** |
| Metric3D V2 | Mono | 2.972 | 0.221 | 0.085 | 0.594 | 0.892 | 0.953 |
| DAC | Mono | 2.915 | 0.229 | 0.087 | 0.590 | 0.889 | 0.953 |
| Ours | DFF | **1.549** | **0.128** | **0.047** | **0.832** | **0.957** | **0.958** |

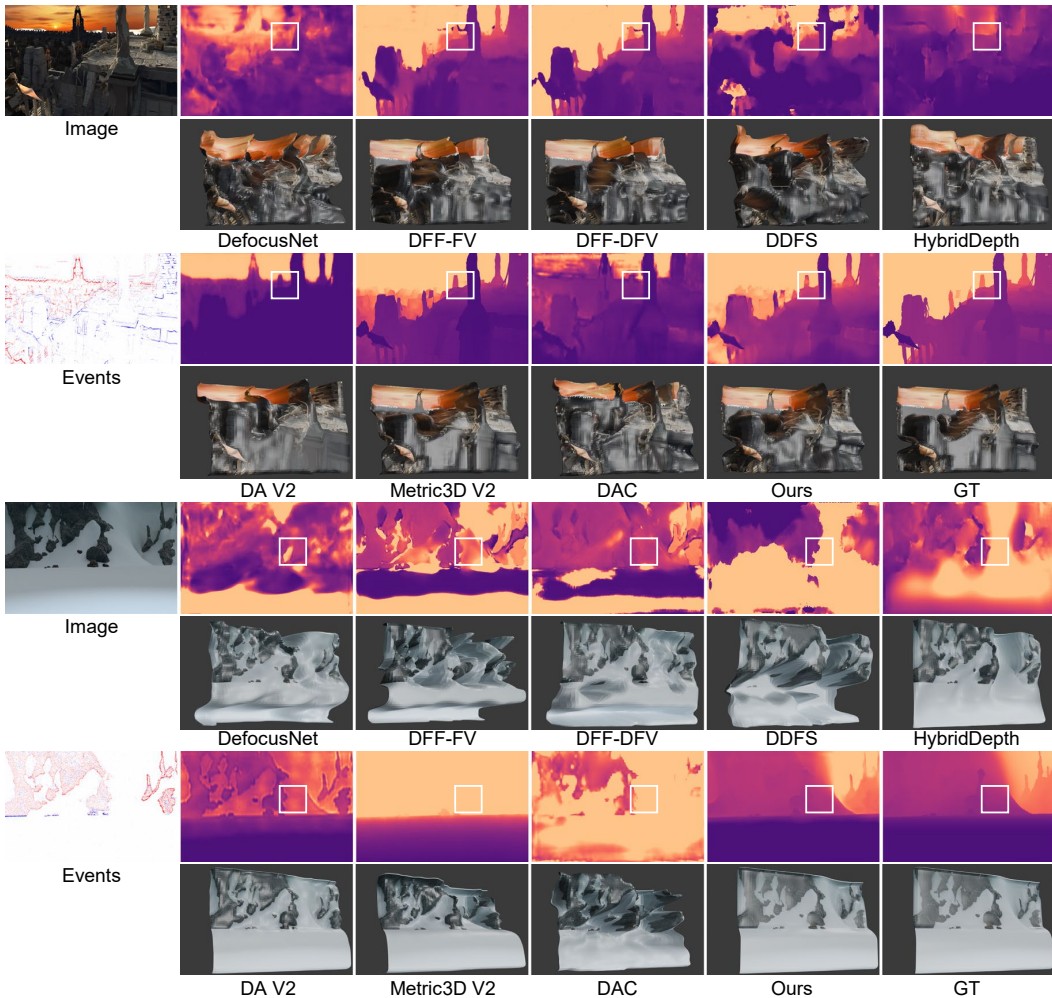

Figure 9: More qualitative results of in-domain experiments on Sintel-Dr. Bokeh dataset with the input image and events.

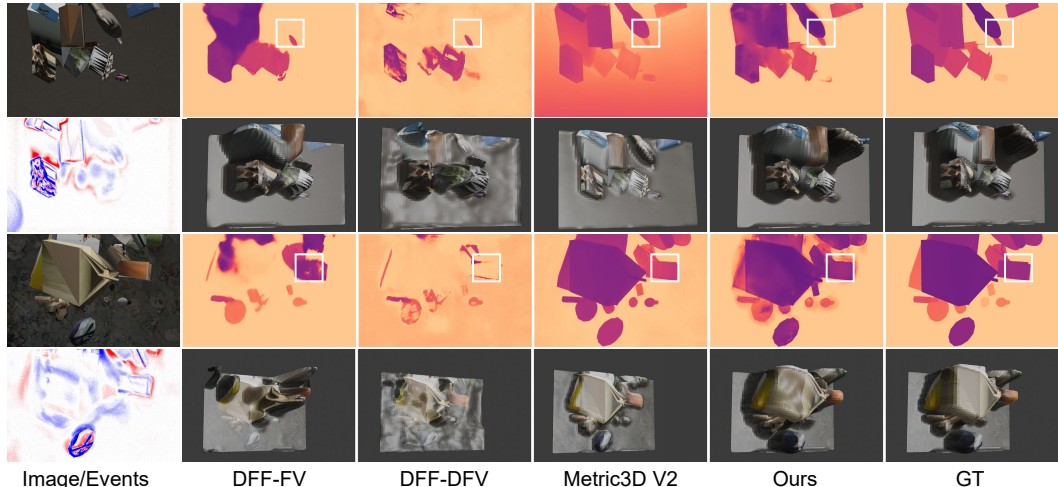

Figure 10: Qualitative results of zero-shot experiments on Blender-Syn dataset with the input image and events. We compare with DFF-FV [64], DFF-DFV [64], and Metric3D V2 [23].

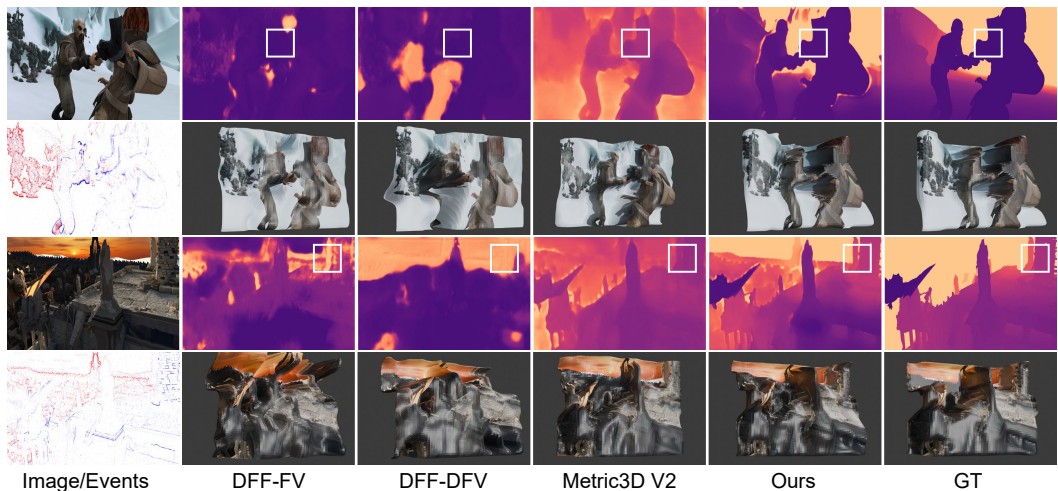

Figure 11: Qualitative results of zero-shot experiments on Sintel-Dr. Bokeh dataset with the input image and events.

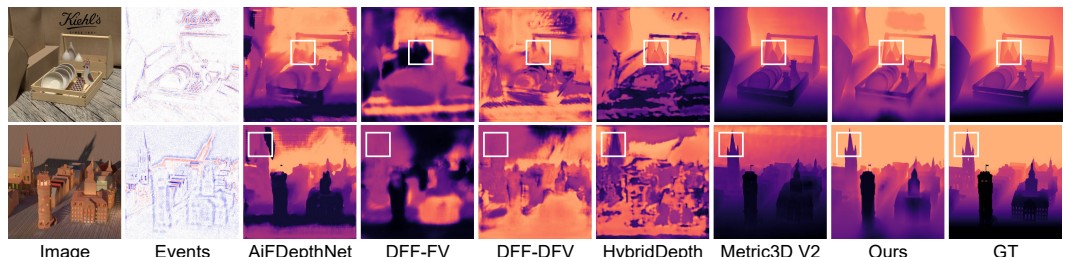

Figure 12: Qualitative results of zero-shot experiments on 4DLFD-Semi-Real dataset with the input image and events. We compare with DFF methods AiFDepthNet [62], DFF-FV, DFF-DFV [64], HybridDepth [14], and single-image-based Metric3D V2 [23].

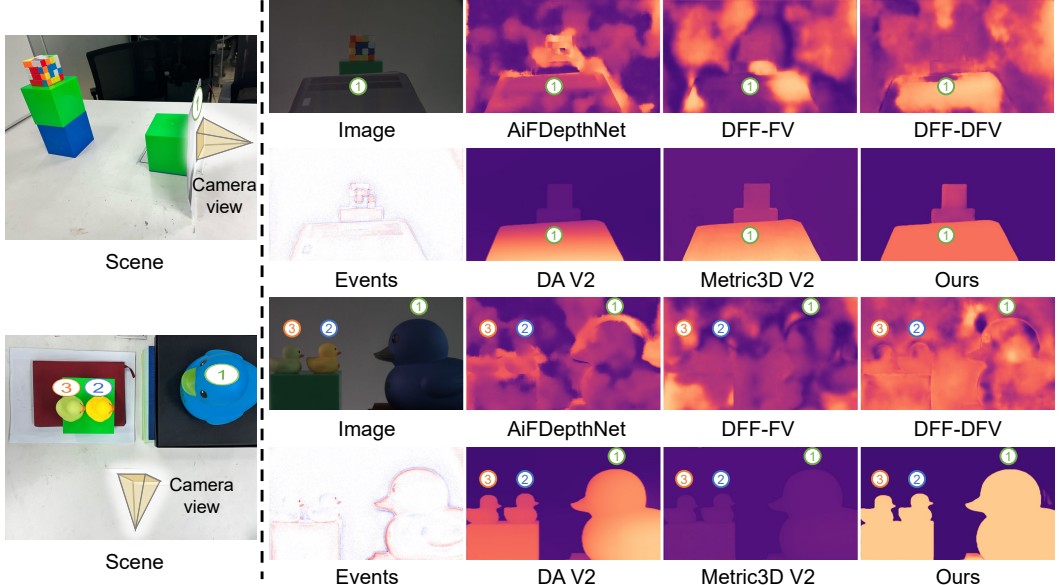

Figure 13: Qualitative results of zero-shot experiments on EDFV-Real dataset. Each pair of consecutive rows corresponds to one test scene. The left side of the dashed line presents the scene overview, with the cones representing camera views, while the right side displays the input along with the outputs of various methods. Each number in the same scene (*i.e.*, ①, ② and ③) corresponds to the same object. Inverse depths are visualized here. Details are in the text.

