# OpenReview forum: "Dense Metric Depth Estimation via Event-based Differential Focus Volume Prompting"
_NeurIPS.cc/2025/Conference — NeurIPS 2025 poster_

### Official Review · Reviewer_1JE8 · 2025-06-16

**Clarity:** 3
**Significance:** 3
**Originality:** 3
**Rating:** 4
**Confidence:** 2

**Summary:**

This paper proposes a novel approach for dense metric depth estimation. In details, the proposed approach intergrades the advantages from both event and RGB images to boost the performance of dense metric depth estimation. The event images are sent to EDFV encoder to obatin EDFV features to obtain sparse depth, while the single view RGB images are passed to a pre-trained image foundation model to get the initial dense depth prediction. A sparseNet and a DenseNet are further used to reflect both outputs into feature space. FInally, a Multi-scale Cross-attention-guided Prompting Module is introudced to fuse the image and event features to obtain the final prediction results. To evaluate the proposed methods, the authors further generate and collect two synthetic datasets and two real-world datasets, and propose the first benchmark for the proposed datasets.

**Questions:**

1. The authors should highlight their contributions on in the framework, as it currently appears that they have merely assembled a set of existing modules.
2. The authors should discuss the implications of using a foundation model, such as the fairness of comparisons with other baselines and the potential risk of information leakage in the zero-shot setting.

**Ethical Concerns:**

["NO or VERY MINOR ethics concerns only"]

**Final Justification:**

Before rebuttal, there are two mainly concerns about my score: 1) The novelty of the proposed method. 2) Experimental setting and information leakage. In the rebuttal, the authors highlight their contributions with more details, which convinces me that their framework design is indeed novel enough for publication. And additional experimental results have also alleviated my concerns about the unfair comparison, so I have revised my rating accordingly. However, regarding the potential information leakage in the zero-shot setting, since I am not particularly familiar with this area and had explicitly indicated a confidence score of 2, I cannot be fully certain about the authors' response. Therefore, I ultimately gave a rating of 4.

**Limitations:**

Yes

**Quality:**

3

**Strengths And Weaknesses:**

Pro:
1. The proposed idea is very interesting and insightful. This pioneering work can serve as an excellent inspiration for the community.
2. The authors conduct extensive experiments to prove the effectiveness of the proposed methods.
3. The newly collected dataset would be a big contribution for the development of the focused research direction.

Cons:
1. Although the proposed idea is very interesting, the proposed framework seems lack of novelty. The main part of the proposed EDPN is the fusion model, which utlizes a cross attention block and a fusion block. However, both cross-attention fusion and conv fusion are all widely used methods in the field of AI. The rest of the proposed framework are all directly rely on existing methods.
2. Different from existing methods, the proposed approach utlizes a pre-trained image foundation model as the image encoder, which seems unfair compared to the other methods. The authors should prove the performance gain is not caused by this.
3. The authors use a pre-trained visual foundation model in zero-shot learning task which may causes information leak. The improvment may leads by this.

Overall, I believe this paper is more suitable for the Datasets & Benchmarks track rather than the Main track.

---

> ### Author Rebuttal · Authors · 2025-07-31
>
> We sincerely thank you for your valuable comments and constructive suggestions! We are encouraged that you find our idea insightful, our experiments extensive and our dataset valuable for the community. Below, we will address your proposed Weaknesses (**W**) and Questions (**Q**) in detail.
> > **W1**: *The proposed framework seems lack of novelty. The main part of the proposed EDPN is the fusion model, which utlizes a cross attention block and a fusion block. However, both cross-attention fusion and conv fusion are all widely used methods in the field of AI. The rest of the proposed framework are all directly rely on existing methods.*
>
> We appreciate your feedback and understand this concern. We respectfully argue that the novelty of our work should be viewed from the **holistic** perspective of the proposed framework (please refer to the proposed second contribution in our paper L63-65). We are the **first** to introduce an EDFV-based prompting method for depth estimation to tackle the problem of scale ambiguities and visual illusions in image foundation models, and experimental results (Sec. 5.3 and 5.4 of our main paper and Sec. E of Supplementary Material) have shown our superiority against foundation models and conventional DFF methods. Each component in the pipeline is essential to the overall method (as discussed in **Ablation studies** in Supplementary Material Sec. D).
>
> While a single component may seem straightforward, our combined pipeline introduces a cohesive and tailored framework that enables effective prompting in a challenging hybrid modality. As discussed in the **Introduction (L48-50)**, the information given by event-based sensors is **different** from other sensors such as traiditional RGB sensors and LiDAR, as it mainly triggers at texture and edge regions. This poses unique challenges and opportunities for our design. In our pipeline, we first utilize proposed EDFV to process events into depth map. After that, we design a SAME strategy to extract effective information from events (**L179-181** of our paper). Besides, considering the modality discrepencies between events and image, we employ different layers to capture their features respectively (**L199-201** of our paper). We also introduce deformable convolutions to handle the sparsity and possible misalignments between two modalities (**L207-209** of our paper). While we agree that the cross-attention fusion and conv fusion may be widely used methods in the field of AI, our adaptation of these modules within a prompting framework tailored for event-based DFF is novel in its use case and validated through extensive experiments. **We will revise the paper to clarify this contribution and better differentiate it from related works.**
>
> > **W2**: *The proposed approach utlizes a pre-trained image foundation model as the image encoder, which seems unfair compared to the other methods.  The authors should prove the performance gain is not caused by this.*
>
> Thank you for raising this point. Although other traditional DFF methods (including DefocusNet [3], DFF-FV [4], DFF-DFV [4], and DDFS [5]) **do not** utilize existing image foundation models, Depth Anything V2 [1] and Metric3D V2 [2] **are foundation models themselves**. The results of DA V2 in Table 1&2 of main paper are finetuned from its pretrained ViT-L checkpoint, while ours **also** utilizes the pretrained DA V2 ViT-L checkpoint as image encoder. Therefore, the performance gain of ours against the finetuned DA V2 should **not** be attributed to the pretrained foundation model, but rather the adaptation of the proposed EDFV and prompting framework.
>
> Besides, in order to exclude the influence of pretrained checkpoints, we further conduct ablation studies on the Blender-Syn dataset. We train DA V2 from scratch on the Blender-Syn dataset **without any pretrained checkpoint**. Then we adopt this DA V2 as the image encoder to train our own framework. The results are shown in the following table, where DA V2 trained from scratch is denoted as "DA V2*", and our method trained with this scheme is denoted as "Ours*". It can be found that even without any pretrained checkpoint, our method still outperforms most of the existing DFF methods in most metrics. Therefore, we believe that the performance gain is **not** caused solely by pretrained image foundation models.
> # Table R1: Quantitative comparisons of in-domain metric depth estimation.
> ## Blender-Syn
> Method|Type|RMSE(↓)|AbsRel(↓)|log10(↓)|$\delta_1$(↑)|$\delta_2$(↑)|$\delta_3$(↑)
> -|-|-|-|-|-|-|-
> DefocusNet|DFF|0.243|0.372|0.107|0.734|0.818|0.861
> DFF-FV|DFF|0.184|0.223|0.062|0.862|0.907|0.926
> DFF-DFV|DFF|0.186|0.250|0.062|$\underline{0.871}$|0.906|0.923
> DDFS|DFF|0.244|0.387|0.109|0.723|0.804|0.849
> DA V2*|Mono|0.149|0.301|0.085|0.738|0.847|0.905
> DA V2|Mono|$\textbf{0.063}$|$\underline{0.089}$|$\underline{0.035}$|0.865|$\underline{0.956}$|$\textbf{0.989}$
> Metric3D V2|Mono|0.095|0.162|0.062|0.826|0.934|0.973
> Ours*|DFF|0.126|0.223|0.058|0.864|0.914|0.938
> Ours|DFF|$\underline{0.068}$|$\textbf{0.077}$|$\textbf{0.028}$|$\textbf{0.919}$|$\textbf{0.972}$|$\underline{0.987}$
>
> One of our contributions is to fuse image foundation models with event-based DFF through our EDFV-based prompting strategy. Therefore, from the comparison with the image foundation models (DA V2 [1] and Metric3D V2 [2]), we demonstrate that our strategy boosts performance, even outperforming these advanced pretrained models. This shows the benefit of our proposed EDFV and fusion framework, beyond the strength of the encoder itself.
>
> > **W3**: *The authors use a pre-trained visual foundation model in zero-shot learning task which may causes information leak.*
>
> Thank you for this important concern. We will address this concern in the following two aspects:
>
> - We have carefully verified the training datasets of our utilized pretrained foundation models, including DA V2 [1] and Metric3D V2 [2], and confirm that **none** of the image datasets used in our evaluations — specifically Sintel and 4D Light Field Dataset — were included in their training corpora.
>
> - We validate generalization performance on **our captured EDFV-real** dataset, as shown in Fig. 5 of main paper and Fig. 12 of Supplementary Material. These results corroborate that our framework performs well in real, unseen scenarios, which could not be attributed to information leakage of foundation models.
>
> > **W4**: *This paper is more suitable for the Datasets & Benchmarks track rather than the Main track.*
>
> We respectfully argue that although the constructed datasets are one of the contributions of our paper, the proposed EDFV representation as well as the prompting framework are also our key contributions. **Reviewer CTBx** has highlighted the novelty of EDFV, and **Reviewer kC23** has emphasized our innovative framework. As far as we know, we are the first work to solve event-based DFF with prompting approach. Please refer to our responses in **W1** for our contributions to the pipeline. Thanks to our EDFV and prompting framework, we could achieve better results than the existing foundation models on both synthetic and real-world datasets (please refer to Sec. 5.3 and 5.4 of our main paper). Therefore, we believe this work aligns well with the **Main track**.
>
> > **Q1**: *The authors should highlight their contributions on in the framework, as it currently appears that they have merely assembled a set of existing modules.*
>
> Please refer to our responses in **W1** and **W4**, where we elaborate on the integration of EDFV, modality-aware fusion, and task-specific prompting — together forming a task-tailored framework.
>
> > **Q2**: *The authors should discuss the implications of using a foundation model, such as the fairness of comparisons with other baselines and the potential risk of information leakage in the zero-shot setting.*
>
> Please refer to our detailed discussions in **W2** and **W3** for the comparison fairness and information leakage.
>
> [1] Yang et al. Depth Anything V2. NeurIPS 2024.
>
> [2] Hu et al. Metric3d V2: A versatile monocular geometric foundation model for zero-shot metric depth and surface normal estimation. TPAMI 2024.
>
> [3] Maximov et al.  Focus on defocus: Bridging the synthetic to real domain gap for depth estimation. CVPR 2020.
>
> [4] Yang et al. Deep Depth from Focus with Differential Focus Volume. CVPR 2022.
>
> [5] Fujimura et al.  Deep depth from focal stack with defocus model for camera-setting invariance. IJCV 2023.

---

> > ### Comment · Reviewer_1JE8 · 2025-08-04
> >
> > Thanks for the response. After reading the rebuttal, I think the authors have addressed all my concerns. Therefore, I'm willing to raise my score.

---

> ### Author Response · Authors · 2025-08-04
> **Thanks for your response and positive feedback**
>
> We sincerely thank you for taking the time to re-evaluate our submission and the positive feedback. We greatly appreciate your thoughtful comments throughout the review process, which helped us improve the quality and clarity of our work. We are glad to hear that our responses addressed your concerns and that you are willing to raise your score.
> Thank you again for your support and encouragement !

---

### Official Review · Reviewer_kC23 · 2025-06-18

**Clarity:** 2
**Significance:** 2
**Originality:** 3
**Rating:** 3
**Confidence:** 3

**Summary:**

This paper introduces a novel approach for dense metric depth estimation by fusing events with image foundation models (IFMs) via a prompting strategy.
The authors propose Event-based Differential Focus Volumes (EDFVs) using events triggered by focus sweeping, transforming them into sparse depth maps to prompt dense estimation via the Event-based Depth Prompting Network (EDPN) and construct synthetic and real datasets for training and evaluation. Experiments show the method outperforms state-of-the-art approaches, achieving up to 29.1% improvement in RMSE.

**Questions:**

Please see 2) and 3) in weakness.

**Ethical Concerns:**

["NO or VERY MINOR ethics concerns only"]

**Final Justification:**

I noticed that more than one reviewer pointed out issues regarding the allocation of space in the paper, such as the absence of ablation studies in the main text and limitations in the comparative experiments. After considering the authors' rebuttal and the feedback from other reviewers, I still think the paper falls slightly below the NeurIPS acceptance threshold. I will maintain my original score. I encourage the authors to revise the paper’s structure and experimental presentation in future versions.

**Limitations:**

Please see the weakness.

**Paper Formatting Concerns:**

No paper Formatting issues.

**Quality:**

2

**Strengths And Weaknesses:**

Strengths:
1. Innovative Modality Fusion: Proposes a novel framework fusing event cameras with image foundation models (IFMs), addressing scale ambiguity in single-image methods and low sampling rates in traditional depth-from-focus approaches.
2. Comprehensive Dataset Construction: Creates synthetic (Blender-Syn, Sintel-Dr. Bokeh) and real-world datasets (4DLFD-Semi-Real, EDFV-Real) with ground truth, enabling rigorous evaluation of frame-based and event-based methods.
3. Superior Performance Validation: Achieves up to 29.1% RMSE improvement over state-of-the-art methods, excelling in in-domain and zero-shot scenarios for textureless regions, scale ambiguity, and occlusion.


Weakness:
1. The organization of the paper could be improved. Ablation studies, as an essential part of the experimental analysis, should be presented in the main text rather than being relegated to the supplementary material. Moreover, since the construction of the two datasets is highlighted as one of the core contributions in the Introduction, a more detailed description of the dataset creation process should also be included in the main body of the paper, rather than placing the majority of this information in the supplementary material.
2. The comparative analysis lacks baseline methods from 2025, which may limit the completeness and timeliness of the evaluation.
3. The experimental section in the main text appears limited in scope. There are no quantitative results reported for the 4DLFD-Semi-Real setting or the proposed dataset. Additionally, the set of comparison models does not include the event-based DFF methods mentioned in the Related Work section.

---

> ### Author Rebuttal · Authors · 2025-07-31
>
> We sincerely thank you for your valuable feedback and constructive suggestions! We are encouraged that you find our framework novel, dataset construction comprehensive and performance superior. Below, we will address your proposed Weaknesses (**W**) in detail.
> > **W1**: *The organization of the paper could be improved. Ablation studies and  a more detailed description of the dataset creation process should be placed in the main paper.*
>
> Thank you for your suggestions! Due to the page limit of the submitted version for peer review, we have to put some details in the Supplementary Material. However, we understand the importance of these components in demonstrating the validity and interpretability of our proposed approach, and will include the important details in the final version of our main paper if accepted as follows:
>
> - We will put **Additional Observation 1&2** (Supplementary Material Sec. A) and their **brief derivations** in the main paper (Sec. 3.2), while retaining the extended math in the Supplement.
>
> - Since the construction of the datasets is one of our core contributions, we will migrate **more dataset construction details** (Supplementary Material Sec. B) to the main paper (Sec. 5.1), including the camera parameters and scene setup details.
>
> - We will move the **ablation studies** (Supplementary Material Sec. D and Table 5) to the main paper as **Sec. 5.5**, following Sec.5.4.
>
> We will condense our texts to make sure the final version of our main paper within page limit, concise, and more inclusive.
> > **W2**: *The comparative analysis lacks baseline methods from 2025.*
>
> Thank you for highlighting this. We have now incorporated a new method, namely **Depth Any Camera (DAC)** [1], published in 2025 to further extend the scope of compared methods. Here we finetune DAC using its pretrained outdoor ResNet101 checkpoint. Please refer to the following tables for our comparison, where we denote DFF methods as "DFF" and monocluar methods as "Mono". We will update the Table 1 and 2 in our final version accordingly. Note that DAC may not perform well because its main focus is wide-angle cameras.
> # Table R1: Quantitative comparisons of in-domain metric depth estimation.
> ## Blender-Syn
> Method|Type|RMSE(↓)|AbsRel(↓)|log10(↓)|$\delta_1$(↑)|$\delta_2$(↑)|$\delta_3$(↑)
> -|-|-|-|-|-|-|-
> DefocusNet|DFF|0.243|0.372|0.107|0.734|0.818|0.861
> DFF-FV|DFF|0.184|0.223|0.062|0.862|0.907|0.926
> DFF-DFV|DFF|0.186|0.250|0.062|$\underline{0.871}$|0.906|0.923
> DDFS|DFF|0.244|0.387|0.109|0.723|0.804|0.849
> DA V2|Mono|$\textbf{0.063}$|$\underline{0.089}$|$\underline{0.035}$|0.865|$\underline{0.956}$|$\textbf{0.989}$
> Metric3D V2|Mono|0.095|0.162|0.062|0.826|0.934|0.973
> DAC|Mono|0.176|0.238|0.115|0.654|0.868|0.947
> Ours|DFF|$\underline{0.068}$|$\textbf{0.077}$|$\textbf{0.028}$|$\textbf{0.919}$|$\textbf{0.972}$|$\underline{0.987}$
> ## Sintel-Dr. Bokeh
> Method|Type|RMSE(↓)|AbsRel(↓)|log10(↓)|$\delta_1$(↑)|$\delta_2$(↑)|$\delta_3$(↑)
> -|-|-|-|-|-|-|-
> DefocusNet|DFF|0.209|0.797|0.192|0.412|0.644|0.728
> DFF-FV|DFF|0.160|0.661|$\underline{0.109}$|$\underline{0.766}$|$\underline{0.863}$|0.898
> DFF-DFV|DFF|$\underline{0.134}$|0.569|$\underline{0.109}$|0.738|0.861|$\underline{0.907}$
> DDFS|DFF|0.282|1.072|0.282|0.441|0.578|0.648
> DA V2|Mono|0.297|0.482|0.361|0.330|0.419|0.472
> Metric3D V2|Mono|0.170|$\underline{0.479}$|0.174|0.452|0.561|0.754
> DAC|Mono|0.273|0.951|0.289|0.268|0.409|0.573
> Ours|DFF|$\textbf{0.095}$|$\textbf{0.141}$|$\textbf{0.072}$|$\textbf{0.806}$|$\textbf{0.901}$|$\textbf{0.945}$
> # Table R2: Quantitative comparisons of zero-shot metric depth estimation.
> ## Blender-Syn
> Method|Type|RMSE(↓)|RMSE log(↓)|log10(↓)|$\delta_1$(↑)|$\delta_2$(↑)|$\delta_3$(↑)
> -|-|-|-|-|-|-|-
> DefocusNet|DFF|0.425|0.783|0.292|0.135|0.391|0.608
> DFF-FV|DFF|0.325|0.661|$\underline{0.162}$|$\underline{0.669}$|$\underline{0.732}$|0.775
> DFF-DFV|DFF|0.369|0.710|0.196|0.651|0.681|0.707
> DDFS|DFF|0.495|1.120|0.377|0.287|0.361|0.448
> DA V2|Mono|0.725|1.913|0.783|0.018|0.057|0.108
> Metric3D V2|Mono|$\underline{0.294}$|$\underline{0.535}$|0.207|0.343|0.625|$\underline{0.777}$
> DAC|Mono|0.652|1.488|0.594|0.075|0.142|0.198
> Ours|DFF|$\textbf{0.148}$|$\textbf{0.282}$|$\textbf{0.081}$|$\textbf{0.697}$|$\textbf{0.878}$|$\textbf{0.944}$
> ## Sintel-Dr. Bokeh
> Method|Type|RMSE(↓)|RMSE log(↓)|log10(↓)|$\delta_1$(↑)|$\delta_2$(↑)|$\delta_3$(↑)
> -|-|-|-|-|-|-|-
> DefocusNet|DFF|0.518|1.504|0.585|0.123|0.204|0.275
> DFF-FV|DFF|$\underline{0.267}$|$\underline{0.982}$|$\underline{0.343}$|0.177|0.364|$\underline{0.518}$
> DFF-DFV|DFF|0.270|1.038|0.366|0.192|0.332|0.495
> DDFS|DFF|0.706|1.852|0.726|0.203|0.231|0.251
> DA V2|Mono|0.337|1.048|0.396|0.242|$\underline{0.391}$|0.461
> Metric3D V2|Mono|0.322|1.174|0.469|$\underline{0.262}$|0.369|0.466
> DAC|Mono|0.515|1.474|0.581|0.144|0.228|0.285
> Ours|DFF|$\textbf{0.233}$|$\textbf{0.685}$|$\textbf{0.253}$|$\textbf{0.333}$|$\textbf{0.466}$|$\textbf{0.560}$
> > **W3.1**: *There are no quantitative results reported for the 4DLFD-Semi-Real setting or the proposed dataset.*
>
> We now provide the **quantitative results** on the 4DLFD-Semi-Real dataset in the following table. Here we convert all disparity predictions into depth values with unified curves for fair comparisons. It can be found that our method outperforms others in all metrics.
> # Table R3: Quantitative comparisons of zero-shot metric depth estimation on 4DLFD-Semi-Real.
> Method|Type|RMSE(↓)|RMSE log(↓)|log10(↓)|$\delta_1$(↑)|$\delta_2$(↑)|$\delta_3$(↑)
> -|-|-|-|-|-|-|-
> DFF-FV|DFF|1.979|0.198|0.070|0.680|0.888|0.949
> DFF-DFV|DFF|1.943|0.186|0.064|0.711|0.902|0.953
> DDFS|DFF|$\underline{1.680}$|$\underline{0.167}$|$\underline{0.060}$|$\underline{0.772}$|$\underline{0.918}$|$\underline{0.956}$
> Metric3D V2|Mono|2.972|0.221|0.085|0.594|0.892|0.953
> DAC|Mono|2.915|0.229|0.087|0.590|0.889|0.953
> Ours|DFF|$\textbf{1.549}$|$\textbf{0.128}$|$\textbf{0.047}$|$\textbf{0.832}$|$\textbf{0.957}$|$\textbf{0.958}$
>
> For EDFV-Real, due to the absence of a depth sensor in our current system setup, we were unable to collect ground-truth depth maps for quantitative evaluation. Please refer to Fig. 5 of our paper and Fig. 12 of Supplementary Material for qualitative comparisons. In the future work, we aim to enhance our hardware system by integrating a depth sensor for accurate ground-truth depth capture, enabling quantitative analysis.
> > **W3.2**: *The set of comparison models does not include the event-based DFF methods mentioned in the Related Work section.*
>
> Thank you for raising this point. The event-based DFF method mentioned in the Related Work section [2] could only predict sparse depth maps. We have contacted its authors, but their code is not yet ready for public release. Besides, we have also found a recent relevent method [3] published in WACV 2025, which leverages existing grayscale video reconstruction approaches for event-based depth estimation. Their method lacks the input image modality, which could constrain their zero-shot prediction performance. Their code is also not released yet, and our email is not responded as of the rebuttal deadline. We will add the comparisons of these methods as soon as they become available before camera ready if accepted.
>
> [1] Guo et al. Depth Any Camera: Zero-Shot Metric Depth Estimation from Any Camera. CVPR 2025.
>
> [2] Jiang et al.  Learning depth from focus with event focal stack. ISJ 2024.
>
> [3] Horikawa et al. Dense Depth from Event Focal Stack. WACV 2025.

---

> > ### Comment · Reviewer_kC23 · 2025-08-04
> > **Reply to authors.**
> >
> > Thank you to the authors for the response and the additional experiments. I noticed that more than one reviewer pointed out issues regarding the allocation of space in the paper, such as the absence of ablation studies in the main text and limitations in the comparative experiments. After considering the authors' rebuttal and the feedback from other reviewers, I still think the paper falls slightly below the NeurIPS acceptance threshold. I will maintain my original score. I encourage the authors to revise the paper’s structure and experimental presentation in future versions. Best of luck.

---

> > > ### Author Response · Authors · 2025-08-04
> > > **Thanks for your response**
> > >
> > > We sincerely thank you for your thoughtful feedback and for taking the time to reconsider our response and additional experiments.
> > > We acknowledge the concern regarding the allocation of space in the original submission. Due to strict page limits, we prioritized presenting our core methodology and key results, which unfortunately led to the omission or relegation of some ablation studies and comparative analyses to the supplementary material. Based on your and other reviewers’ helpful comments, we will move more mathematical derivations, dataset construction details and ablation studies to the main paper in the camera-ready version if accepted. We also plan to improve the clarity of our text to better support our claims.
> > > We truly appreciate your constructive suggestions, which is invaluable for this work. Thank you again for your time and for encouraging us to strengthen the paper.

---

### Official Review · Reviewer_hHyQ · 2025-07-01

**Clarity:** 3
**Significance:** 2
**Originality:** 3
**Rating:** 4
**Confidence:** 4

**Summary:**

This paper proposes to fuse events with depth foundation models for metric depth estimation. The main contributions include 1) a EDFV representation that encodes focus sweeping cues from events. 2) a framework that combines events and depth foundation model for metric depth estimation. 3) novel datasets for proposed task. Experimental results on Blender-Syn and Sintel-Dr. Bokeh datasets demonstrate the superiority of proposed method.

**Questions:**

- The necessity of using events for focus/defocus cue encoding is not sufficiently justified. The data collection process still relies on focal length changing, and the authors note that event cameras are used to record focal length changes more frequently. However, since focal length changes are relatively slow, it is unclear whether such frequent recording is necessary, as standard cameras can capture more than 30 frames per second during focal length changes. This raises questions about whether a focal stack constructed from standard cameras is sufficient for accurate depth estimation. Additionally, events triggered by factors other than focal length changes may introduce noise, potentially degrading depth estimation performance. The authors should provide further analysis to demonstrate the superiority of using event cameras over traditional depth-from-focus methods.
- Based on the above question, if we replace the sparse depth generation step with normal depth-from-focus approaches, and keep the subsequent steps (MCPM part) unchanged, how would the performance be affected?
-  It is suggested to indicate the type of compared method (such as DFF and Mono) in Table 1 and 2 for better clarity.

**Ethical Concerns:**

["NO or VERY MINOR ethics concerns only"]

**Final Justification:**

Please see the comment.

**Limitations:**

The limitation is discussed in the main text.

**Paper Formatting Concerns:**

There is no formatting concern.

**Quality:**

3

**Strengths And Weaknesses:**

Strength
- The exploration of encoding focus/defocus cues from events addresses an underexplored task in the field.
- The proposed method demonstrates superior performance compared to existing approaches.
- The introduction of new datasets is a valuable contribution to the community.

Weakness
- The frequent use of abbreviations throughout the paper may confuse readers, reducing clarity.
- The paper’s structure needs improvement. Notably, the Ablation Study section is recommended to put in the main text.
- The experimental evaluation lacks comparisons with state-of-the-art approaches from the past two years, which weakens the claim of superior performance.
- Although the 4DLFD-Semi-Real and EDFV-Real datasets are introduced, no quantitative experiments are conducted on these datasets.
- Some typo: L197, "we two".

---

> ### Author Rebuttal · Authors · 2025-07-31
>
> We sincerely thank you for your insightful comments and constructive suggestions! We are encouraged that you find our idea novel, our performance superior, and our dataset valuable to the community. Below, we will address your proposed Weaknesses (**W**) and Questions (**Q**) in detail.
> > **W1**: *The frequent use of abbreviations throughout the paper may confuse readers.*
>
> Thank you for your feedback. We will revise the entire manuscript to minimize the use of unnecessary abbreviations, and ensure they are clearly defined at their first appearance in the main text to improve readability.
> > **W2**: *The paper’s structure needs improvement. The Ablation Study section is recommended to put in the main text.*
>
> Thank you for your suggestions! Due to the page limit of the submitted version for peer review, we have to put some details in the Supplementary Material. However, we understand the importance of these components in demonstrating the validity and interpretability of our proposed approach, and will include the important details in the final version of our main paper if accepted as follows:
>
> - We will put **Additional Observation 1&2** (Supplementary Material Sec. A) and their **brief derivations** in the main paper (Sec. 3.2), while retaining the extended math in the Supplement.
>
> - Since the construction of the datasets is one of our core contributions, we will migrate **more dataset construction details** (Supplementary Material Sec. B) to the main paper (Sec. 5.1), including the camera parameters and scene setup details.
>
> - We will move the **ablation studies** (Supplementary Material Sec. D and Table 5) to the main paper as **Sec. 5.5**, following Sec.5.4.
>
> We will condense our texts to make sure the final version of our main paper within page limit, concise, and more inclusive.
>
> > **W3**: *The experimental evaluation lacks comparisons with state-of-the-art approaches from the past two years.*
>
> We respectfully clarify that **Depth Anything V2** [1] and **Metric3D V2** [2] are both state-of-the-art depth estimation methods published in the year 2024. In addition, we have now incorporated a new method, namely **Depth Any Camera (DAC)** [3], published in 2025 to further extend the scope of compared methods. Here we finetune DAC using its pretrained outdoor ResNet101 checkpoint. Please refer to the following tables for our comparison, where we denote DFF methods as "DFF" and monocluar methods as "Mono". We will update the Table 1 and 2 in our final version accordingly. Note that DAC may not perform well because its main focus is wide-angle cameras.
> # Table R1: Quantitative comparisons of in-domain metric depth estimation.
> ## Blender-Syn
> Method|Type|RMSE(↓)|AbsRel(↓)|log10(↓)|$\delta_1$(↑)|$\delta_2$(↑)|$\delta_3$(↑)
> -|-|-|-|-|-|-|-
> DefocusNet|DFF|0.243|0.372|0.107|0.734|0.818|0.861
> DFF-FV|DFF|0.184|0.223|0.062|0.862|0.907|0.926
> DFF-DFV|DFF|0.186|0.250|0.062|$\underline{0.871}$|0.906|0.923
> DDFS|DFF|0.244|0.387|0.109|0.723|0.804|0.849
> DA V2|Mono|$\textbf{0.063}$|$\underline{0.089}$|$\underline{0.035}$|0.865|$\underline{0.956}$|$\textbf{0.989}$
> Metric3D V2|Mono|0.095|0.162|0.062|0.826|0.934|0.973
> DAC|Mono|0.176|0.238|0.115|0.654|0.868|0.947
> Ours|DFF|$\underline{0.068}$|$\textbf{0.077}$|$\textbf{0.028}$|$\textbf{0.919}$|$\textbf{0.972}$|$\underline{0.987}$
> ## Sintel-Dr. Bokeh
> Method|Type|RMSE(↓)|AbsRel(↓)|log10(↓)|$\delta_1$(↑)|$\delta_2$(↑)|$\delta_3$(↑)
> -|-|-|-|-|-|-|-
> DefocusNet|DFF|0.209|0.797|0.192|0.412|0.644|0.728
> DFF-FV|DFF|0.160|0.661|$\underline{0.109}$|$\underline{0.766}$|$\underline{0.863}$|0.898
> DFF-DFV|DFF|$\underline{0.134}$|0.569|$\underline{0.109}$|0.738|0.861|$\underline{0.907}$
> DDFS|DFF|0.282|1.072|0.282|0.441|0.578|0.648
> DA V2|Mono|0.297|0.482|0.361|0.330|0.419|0.472
> Metric3D V2|Mono|0.170|$\underline{0.479}$|0.174|0.452|0.561|0.754
> DAC|Mono|0.273|0.951|0.289|0.268|0.409|0.573
> Ours|DFF|$\textbf{0.095}$|$\textbf{0.141}$|$\textbf{0.072}$|$\textbf{0.806}$|$\textbf{0.901}$|$\textbf{0.945}$
> # Table R2: Quantitative comparisons of zero-shot metric depth estimation.
> ## Blender-Syn
> Method|Type|RMSE(↓)|RMSE log(↓)|log10(↓)|$\delta_1$(↑)|$\delta_2$(↑)|$\delta_3$(↑)
> -|-|-|-|-|-|-|-
> DefocusNet|DFF|0.425|0.783|0.292|0.135|0.391|0.608
> DFF-FV|DFF|0.325|0.661|$\underline{0.162}$|$\underline{0.669}$|$\underline{0.732}$|0.775
> DFF-DFV|DFF|0.369|0.710|0.196|0.651|0.681|0.707
> DDFS|DFF|0.495|1.120|0.377|0.287|0.361|0.448
> DA V2|Mono|0.725|1.913|0.783|0.018|0.057|0.108
> Metric3D V2|Mono|$\underline{0.294}$|$\underline{0.535}$|0.207|0.343|0.625|$\underline{0.777}$
> DAC|Mono|0.652|1.488|0.594|0.075|0.142|0.198
> Ours|DFF|$\textbf{0.148}$|$\textbf{0.282}$|$\textbf{0.081}$|$\textbf{0.697}$|$\textbf{0.878}$|$\textbf{0.944}$
> ## Sintel-Dr. Bokeh
> Method|Type|RMSE(↓)|RMSE log(↓)|log10(↓)|$\delta_1$(↑)|$\delta_2$(↑)|$\delta_3$(↑)
> -|-|-|-|-|-|-|-
> DefocusNet|DFF|0.518|1.504|0.585|0.123|0.204|0.275
> DFF-FV|DFF|$\underline{0.267}$|$\underline{0.982}$|$\underline{0.343}$|0.177|0.364|$\underline{0.518}$
> DFF-DFV|DFF|0.270|1.038|0.366|0.192|0.332|0.495
> DDFS|DFF|0.706|1.852|0.726|0.203|0.231|0.251
> DA V2|Mono|0.337|1.048|0.396|0.242|$\underline{0.391}$|0.461
> Metric3D V2|Mono|0.322|1.174|0.469|$\underline{0.262}$|0.369|0.466
> DAC|Mono|0.515|1.474|0.581|0.144|0.228|0.285
> Ours|DFF|$\textbf{0.233}$|$\textbf{0.685}$|$\textbf{0.253}$|$\textbf{0.333}$|$\textbf{0.466}$|$\textbf{0.560}$
> > **W4**: *No quantitative experiments are conducted on 4DLFD-Semi-Real and EDFV-Real datasets.*
>
> We now provide the **quantitative results** on the 4DLFD-Semi-Real dataset in the following table. Here we convert all disparity predictions into depth values with unified curves for fair comparisons. It can be found that our method outperforms others in all metrics.
> # Table R3: Quantitative comparisons of zero-shot metric depth estimation on 4DLFD-Semi-Real.
> Method|Type|RMSE(↓)|RMSE log(↓)|log10(↓)|$\delta_1$(↑)|$\delta_2$(↑)|$\delta_3$(↑)
> -|-|-|-|-|-|-|-
> DFF-FV|DFF|1.979|0.198|0.070|0.680|0.888|0.949
> DFF-DFV|DFF|1.943|0.186|0.064|0.711|0.902|0.953
> DDFS|DFF|$\underline{1.680}$|$\underline{0.167}$|$\underline{0.060}$|$\underline{0.772}$|$\underline{0.918}$|$\underline{0.956}$
> Metric3D V2|Mono|2.972|0.221|0.085|0.594|0.892|0.953
> DAC|Mono|2.915|0.229|0.087|0.590|0.889|0.953
> Ours|DFF|$\textbf{1.549}$|$\textbf{0.128}$|$\textbf{0.047}$|$\textbf{0.832}$|$\textbf{0.957}$|$\textbf{0.958}$
>
> For EDFV-Real, due to the absence of a depth sensor in our current system setup, we were unable to collect ground-truth depth maps for quantitative evaluation. Please refer to Fig. 5 of our paper and Fig. 12 of Supplementary Material for qualitative comparisons. In the future work, we aim to enhance our hardware system by integrating a depth sensor for accurate ground-truth depth capture, enabling quantitative analysis.
> > **W5**: *Some typo: L197, "we two".*
>
> Thank you for spotting this. Here we have left an "employ", which should be "we employ two U-Net-based networks". We will thoroughly revise the manuscript and eliminate all such typographical errors in our final version.
> > **Q1&Q2**: *The authors should provide further analysis to demonstrate the superiority of using event cameras over traditional depth-from-focus methods. Based on the above question, if we replace the sparse depth generation step with normal depth-from-focus approaches, and keep the subsequent steps (MCPM part) unchanged, how would the performance be affected?*
>
> We appreciate this insightful question. We respectfully demonstrate the necessity of introducing events in the following three aspects:
>
> 1. Our current setup is just a **camera prototype**, and the rotation speed is limited by mechanical or manual speed. However, as far as we know, some industrial camera modules could reduce the focus ring rotation time to about **10ms** such as the liquid lens, which would be hard for standard cameras to capture focal stacks. However, event cameras can easily handle the scenario with a **microsecond-level** temporal resolution.
>
> 2. As discussed in **L171-174** of our paper, the performance of traditional DFF methods could be influenced by the **discrete** frame number in focal stacks. However, with events, we can capture the focal stack with a microsecond-level temporal resolution, which can provide **nearly continuous** transitions between in-focus and out-of-focus regions, and help to improve the accuracy of depth estimation. In addition, although events may be triggered less in **smooth or textureless** regions, traditional DFF methods also struggle in such areas [4] as they depend on textures to estimate the in-focus degree. Therefore, even if replacing the sparse depth generation step with normal DFF approaches, the performance **may not increase**.
>
> 3. As events only record intensity changes, the total raw data size with resolution $720\times 1280$ during one capture is averagely **4-5MB**. However, a single image in the focal stack with the same resolution could take up to 2.7M, with 30 images up to **80MB**. This means that the storage space required for event-based depth estimation is much smaller than that of traditional DFF methods, which make it more practical to deploy on resource-limited devices.
> > **Q3**: *It is suggested to indicate the type of compared method (such as DFF and Mono) in Table 1 and 2 for better clarity.*
>
> Thank you for your suggestions. We have provided the updated version of Table 1 and 2 in the tables above, and will also include them in our final version.
>
> [1] Yang et al. Depth Anything V2. NeurIPS 2024.
>
> [2] Hu et al. Metric3d V2: A versatile monocular geometric foundation model for zero-shot metric depth and surface normal estimation. TPAMI 2024.
>
> [3] Guo et al. Depth Any Camera: Zero-Shot Metric Depth Estimation from Any Camera. CVPR 2025.
>
> [4] Yang et al. Deep Depth from Focus with Differential Focus Volume. CVPR 2022.

---

> > ### Comment · Reviewer_hHyQ · 2025-08-01
> >
> > Thank you for the detailed response. Most of my concerns have been addressed. However, regarding W3, my original point was that recent DFF methods were missing from the comparison. For instance, the most recent DFF method, DDFS, was published in 2023. To better demonstrate the effectiveness of the proposed approach, I recommend incorporating comparisons with more recent DFF methods.

---

> > ### Comment · Reviewer_hHyQ · 2025-08-02
> >
> > Furthermore, for the second poin in Q1&Q2, it is suggested to demonstrate the argument through quantitative experiments.

---

> ### Author Response · Authors · 2025-08-04
> **(Updated) Experimental results of a recent DFF method and an ablation study**
>
> Thank you for your responses! We have quickly reviewed related literatures in the past two years and now added a new DFF method for comparison, namely **HybridDepth** [1], which is a recent method published in **WACV 2025**. This method incorporates both a pretrained traditional DFF method (DFF-DFV in their implementation) and a pretrained image foundation model (DA in their implementation) as a relative depth estimator. Although taking information from both sources, it may still face the challenge of discrete frame number and acquisition time discussed in our paper as they need image focal stacks as input. We have now fintuned it on our two synthetic datasets from its pretrained NYU checkpoint and provide the in-domain quantitative results in Table R1 below. HybridDepth generates its output by multiplying the estimation results of DA with a predicted scale map, which may be the reason why it doesn't preform well on Sintel-Dr.bokeh dataset with large scale variations across different scenes. In addition, we have also noticed a **CVPR 2025** paper [2] that utilizes defocus clues for sparse and dense depth estimation. However, the code for dense depth estimation is not available yet, and we will try to include it for comparison once it is available.
>
> Besides, in order to demonstrate our second point in **Q1&Q2**, we further conduct an ablation study, where we have replaced our sparse depth generation network with an existing DFF method (here we use **DFF-DFV**) and keep the rest of our framework unchanged. For fair comparison, we employ 32 frames as the input focal stack, and use the same random initialization and the same training scheme on this network (denoted as **Ours#**). Here we also report the results of this network in the **Table R1** below, and provide the computational efficiency comparison in **Table R2**. It can be found that although Ours# brings **minor** improvements to **some** metrics compared with Ours (such as 0.001 in RMSE and 0.001 in $\delta_1$), it brings **much more** computational complexity (3.42 times parameters and 8.74 times MACs) to the whole pipeline, which is not desirable for resource-limited devices. Instead, our event-based method can achieve a balance between accuracy and efficiency, and is more practical to deploy on resource-limited devices.
>
> Thank you again for your thoughtful comments and the continued discussion, which helps us further improve our work.
> # Table R1: Quantitative comparisons of in-domain metric depth estimation.
> ## Blender-Syn
> Method|Type|RMSE(↓)|AbsRel(↓)|log10(↓)|$\delta_1$(↑)|$\delta_2$(↑)|$\delta_3$(↑)
> -|-|-|-|-|-|-|-
> DefocusNet|DFF|0.243|0.372|0.107|0.734|0.818|0.861
> DFF-FV|DFF|0.184|0.223|0.062|0.862|0.907|0.926
> DFF-DFV|DFF|0.186|0.250|0.062|0.871|0.906|0.923
> DDFS|DFF|0.244|0.387|0.109|0.723|0.804|0.849
> HybridDepth|DFF|0.089|0.123|0.051|0.823|0.925|0.969
> DA V2|Mono|$\textbf{0.063}$|0.089|$\underline{0.035}$|0.865|0.956|$\textbf{0.989}$
> Metric3D V2|Mono|0.095|0.162|0.062|0.826|0.934|0.973
> DAC|Mono|0.176|0.238|0.115|0.654|0.868|0.947
> Ours#|DFF|$\underline{0.067}$|$\underline{0.079}$|$\textbf{0.028}$|$\textbf{0.920}$|$\underline{0.968}$|0.985
> Ours|DFF|0.068|$\textbf{0.077}$|$\textbf{0.028}$|$\underline{0.919}$|$\textbf{0.972}$|$\underline{0.987}$
> ## Sintel-Dr. Bokeh
> Method|Type|RMSE(↓)|AbsRel(↓)|log10(↓)|$\delta_1$(↑)|$\delta_2$(↑)|$\delta_3$(↑)
> -|-|-|-|-|-|-|-
> DefocusNet|DFF|0.209|0.797|0.192|0.412|0.644|0.728
> DFF-FV|DFF|0.160|0.661|$\underline{0.109}$|$\underline{0.766}$|$\underline{0.863}$|0.898
> DFF-DFV|DFF|$\underline{0.134}$|0.569|$\underline{0.109}$|0.738|0.861|$\underline{0.907}$
> DDFS|DFF|0.282|1.072|0.282|0.441|0.578|0.648
> HybridDepth|DFF|0.273|0.657|0.295|0.233|0.393|0.540
> DA V2|Mono|0.297|0.482|0.361|0.330|0.419|0.472
> Metric3D V2|Mono|0.170|$\underline{0.479}$|0.174|0.452|0.561|0.754
> DAC|Mono|0.273|0.951|0.289|0.268|0.409|0.573
> Ours|DFF|$\textbf{0.095}$|$\textbf{0.141}$|$\textbf{0.072}$|$\textbf{0.806}$|$\textbf{0.901}$|$\textbf{0.945}$
> # Table R2: Computational efficiency comparison.
> Method|Params(M)|MACs(G)|Time(ms)
> -|-|-|-
> Ours#|20.41|134.91|28.40
> Ours|$\textbf{5.96}$|$\textbf{15.44}$|$\textbf{17.40}$
>
> [1] Ganj et al. HybridDepth: Robust Depth Fusion by Leveraging Depth from Focus and Single-Image Priors. WACV 2025.
>
> [2] Xu et al. Blurry-Edges: Photon-Limited Depth Estimation from Defocused Boundaries. CVPR 2025.

---

> ### Author Response · Authors · 2025-08-06
> **We have updated our experimental results of the new DFF method**
>
> We sincerely thank you again for your suggestions and further discussions! Now we have completed our training and evaluation of HybridDepth on Sintel-Dr. Bokeh dataset, and the results can be found in the updated Table R1 of of our last Offical Comment for you. It can be found that our method still outperforms HybridDepth in all metrics. HybridDepth generates its output by multiplying the estimation results of DA with a predicted scale map, which may be the reason why it doesn't preform well on Sintel-Dr. Bokeh dataset with large scale variations across different scenes. We will add these comparisons to our final version if accepted. Please let us know if there are more comments and we will be more than happy to follow up.

---

> > ### Comment · Reviewer_hHyQ · 2025-08-07
> >
> > I appreciate the author for their detailed response. All things considered, I will keep my positive rating unchanged.

---

> > > ### Author Response · Authors · 2025-08-07
> > > **Thanks for your response**
> > >
> > > We would like to express our sincere gratitude for your thoughtful consideration of our work and our rebuttal! We are pleased to hear that you found our response detailed and are very thankful for your decision to maintain your positive rating. Your constructive suggestions are very helpful in improving the quality of our work.

---

### Official Review · Reviewer_CTBx · 2025-07-05

**Clarity:** 3
**Significance:** 3
**Originality:** 3
**Rating:** 4
**Confidence:** 4

**Summary:**

This paper proposes a depth estimation framework that combines event camera data generated during focus sweeping with image foundation models via a prompting strategy. It introduces the Event-based Differential Focus Volume (EDFV) for sparse metric depth estimation and designs an Event-based Depth Prompting Network (EDPN) to refine dense depth predictions. The authors also construct synthetic and real datasets to evaluate their method.

1. EDFV (Event-based Differential Focus Volumes) for extracting sparse metric depth from events.
2. EDPN (Event-based Depth Prompting Network) that fuses sparse event depth and dense IFM predictions to produce final accurate depth.
3. Construction of two synthetic datasets and two real/semi-real datasets with events, AIF images, and GT depth for training and evaluation.
4. Superior in-domain and zero-shot performance compared to state-of-the-art DFF and single-image methods across multiple datasets.

**Questions:**

1. While the proposed EDFV performs well overall, it's unclear how it handles textureless regions or areas with low event density. Could the authors show or describe failure cases where EDFV performs poorly? Do you apply any fallback mechanism or fusion weighting in such cases?

2. In your real-world dataset (EDFV-Real), how did you ensure temporal synchronization between the event camera and RGB sensor, especially during fast focus sweeping? Were there any delays or misalignment corrections applied? Accurate depth estimation heavily depends on synchronization.

**Ethical Concerns:**

["NO or VERY MINOR ethics concerns only"]

**Final Justification:**

The authors have addressed my concerns raised during the review process.

**Limitations:**

1. The method relies on event triggers from intensity changes, which may be sparse or absent in textureless or poorly lit regions.

2. The proposed system requires precise synchronization between RGB and event cameras and a controllable focus mechanism. These requirements limit real-world usability, especially for mobile or embedded systems.

**Quality:**

3

**Strengths And Weaknesses:**

Strengths:

Quality :
The proposed method is well-motivated and carefully designed. The derivation of the polarity reversal property in Section 3.2 shows a solid understanding of event camera properties. Extensive experiments are conducted on four datasets (two synthetic, two real/semi-real), with both in-domain and zero-shot evaluations.

Clarity:
The paper is well-organized, with clearly separated contributions and pipeline stages.

Significance:
The integration of event-based sparse depth with foundation model prompting addresses the critical problem of scale ambiguity in monocular depth estimation. The event-guided prompting idea may influence future depth estimation systems or even broader tasks involving hybrid sensing and prompting.

Originality:
The proposal of EDFV is novel in the context of depth estimation, leveraging event-triggered defocus polarity—a concept not previously exploited in this form. The method diverges meaningfully from both traditional DFF and event-based monocular methods. The use of deformable convolutions in cross-attention for misalignment correction is an innovative adaptation to the challenges of hybrid sensor data.

Weaknesses:

Clarity:
Some important details are left to the Supplementary Material (e.g., more mathematical derivation of event polarity behavior, ablation studies on hyperparameters). The limitations section is quite short. A more systematic reflection on assumptions (e.g., sensitivity to event noise, synchronization of sensors) would improve transparency.

Significance:
While promising, the real-world applicability is still partially constrained by the hardware complexity of event-RGB hybrid systems and the assumption of focus sweeping capability. These aspects are not discussed in depth.

Originality:
The idea of prompting IFMs with sparse depth is increasingly common (e.g., in LiDAR prompting literature). Although the paper adapts this idea well to event data, the prompting framework (EDPN) may not be as novel as the EDFV component.

---

> ### Author Rebuttal · Authors · 2025-07-31
>
> We sincerely thank you for your insightful comments and constructive suggestions! We are encouraged that you find our method well-motivated and carefully designed, our paper well-organized, and our concept novel. Below, we provide detailed responses to your proposed Weaknesses (**W**), Questions (**Q**), and Limitations (**L**).
> > **W1.1**: *Some important details are left to the Supplementary Material.*
>
> Thank you for your suggestions! Due to the page limit of the submitted version for peer review, we have to put some details in the Supplementary Material. However, we understand the importance of these components in demonstrating the validity and interpretability of our proposed approach, and will include the important details in the final version of our main paper if accepted as follows:
>
> - We will put **Additional Observation 1&2** (Supplementary Material Sec. A) and their **brief derivations** in the main paper (Sec. 3.2), while retaining the extended math in the Supplement.
>
> - Since the construction of the datasets is one of our core contributions, we will migrate **more dataset construction details** (Supplementary Material Sec. B) to the main paper (Sec. 5.1), including the camera parameters and scene setup details.
>
> - We will move the **ablation studies** (Supplementary Material Sec. D and Table 5) to the main paper as **Sec. 5.5**, following Sec.5.4.
>
> We will condense our texts to make sure the final version of our main paper within page limit, concise, and more inclusive.
> > **W1.2**: *The limitations section is quite short. A more systematic reflection on assumptions (e.g., sensitivity to event noise, synchronization of sensors) would improve transparency.*
>
> We appreciate this suggestion. While our method has shown robustness in various scenarios,  we acknowledge its performance may degrade under extreme conditions. Specifically:
> - In **low light scenarios**, high sensor noise can reduce the quality of EDFV and predicted sparse depth, thus causing the performance of our method to degrade.
> - When applied to **high-speed scenarios**, the synchronization of sensors could be an important issue as we need the frame and events to capture the scene with similar timing. But there are some approaches such as the chip synchronization in DAVIS346 [1] and external clock triggering mechanism [2] to limit the time error within 10ms. Besides, a specifically designed electronic system [6] could reach the time precision of **0.5ms**. These hardware solutions could enable our system to achieve high-precision synchronization.
>
> We will add these discussions in the final version of our paper, and try to improve the robustness of our method in these challenging scenarios in our future work.
> > **W2**: *While promising, the real-world applicability is still partially constrained by the hardware complexity of event-RGB hybrid systems and the assumption of focus sweeping capability. These aspects are not discussed in depth.*
>
> Thank you for highlighting these practical considerations.
> - For the hardware complexity, our current setup is just a **camera prototype**, and the hybrid system could be further integrated by engineering approches to reduce the complexity. In addition, as far as we know, some industrial camera modules could reduce the focus ring rotation time to about **10ms** such as the liquid lens, which could further improve the real-time efficiency of our system.
> - For the assumption of focus sweeping capability, as indicated in the mathematical derivations in Sec. 3.2 and Supplementary Material Sec. A, the effectiveness of events mainly locates at edges and textures. Therefore, for textureless regions or areas with low event density, our estimation relies more heavily on the initial dense depth and sparse depth around textured regions. This may cause our method to perform suboptimally in **highly smooth scenarios**. Notably, such scenarios also remain **open challenges** for conventional DFF approaches [3], as they depend on textures to estimate the in-focus degree.
>
> We will add these discussions in the final version of our paper, and work on improving the robustness of our method in these scenarios in future work.
> > **W3**: *The idea of prompting IFMs with sparse depth is increasingly common (e.g., in LiDAR prompting literature). The prompting framework (EDPN) may not be as novel as the EDFV component.*
>
> Thank you for this thoughtful comment. We agree that the general idea of prompting image foundation models (IFMs) with sparse depth has gained increasing attention, such as in the context of LiDAR [4,5]. However, although **the idea is common, the strategy may differ**.  Therefore, we respectfully argue that our approach differs from previous works in motivation and implementation. It should be noted that the novelty of our work should be viewed from the **holistic** perspective of the proposed framework, compromising the EDFV construction, sparse depth & initial dense depth estimation, and prompting network (EDPN) (please refer to the proposed second contribution in our paper L63-65). Each component in the pipeline is essential to the overall method (as discussed in **Ablation studies** in Supplementary Material Sec. D). While a single component may seem straightforward, our combined pipeline introduces a cohesive and tailored framework that enables effective prompting in a challenging hybrid modality.
>
>  As discussed in the **Introduction (L48-50)**, the information given by event-based sensors is different from other sensors such as LiDAR, as it mainly triggers at texture and edge regions. This poses unique challenges and opportunities for our design.
>
> In our pipeline, we first utilize proposed EDFV to process events into depth map. After that, we design a SAME strategy to extract effective information from events (**L179-181** of our paper). Besides, considering the modality discrepancies between events and image, we employ different layers to capture their features respectively (**L199-201** of our paper). We also introduce deformable convolutions to handle the sparsity and possible misalignments between two modalities (**L207-209** of our paper). **We will revise the paper to clarify this contribution and better differentiate it from related works.**
>
> > **Q1&L1**: *The method relies on event triggers from intensity changes, which may be sparse or absent in textureless or poorly lit regions. Could the authors show or describe failure cases where EDFV performs poorly? Do you apply any fallback mechanism or fusion weighting in such cases?*
>
>  Please refer to our rebuttal in **W1.2** and **W2** for some challenging scenarios where our method may perform suboptimally. As we have considered the feature of sparse events in our design of prompting network, we did not incorporate a fallback mechanism in our method. The deformable convolutions and cross attention mechanism in our pipeline could help to handle the textureless regions. However, we agree that an additional fallback mechanism or fusion weighting may further improve the performance in extremely challenging conditions. These improvements are beyond our current scope, and will be considered as our future work.
> > **Q2&L2**: *The proposed system requires precise synchronization between RGB and event cameras and a controllable focus mechanism. These requirements limit real-world usability. How to ensure temporal synchronization between the event camera and RGB sensor, especially during fast focus sweeping?*
>
> If applied to static or slow motion scenarios, the synchronization between the event camera and RGB sensor may be less crucial, as they record nearly the same scene over short durations. But if applied to high-speed scenarios, tighter synchronization could be required. As mentioned in **W1.2**, hardware-based solutions such as externally triggered clocking mechanisms [2] are for maintaining reliable timing within temporal tolerances.
>
> [1] Zhu et al. The Multi Vehicle Stereo Event Camera Dataset: An Event Camera Dataset for 3D Perception. RAL 2018.
>
> [2] Duan et al. EventAid: Benchmarking Event-aided Image/Video Enhancement Algorithms with Real-captured Hybrid Dataset. TPAMI 2025.
>
> [3] Yang et al. Deep Depth from Focus with Differential Focus Volume. CVPR 2022.
>
> [4] Park et al. Depth Prompting for Sensor-Agnostic Depth Estimation. CVPR 2024.
>
> [5] Lin et al. Prompting Depth Anything for 4K Resolution Accurate Metric Depth Estimation. CVPR 2025.
>
> [6] Zou et al. Learning to Reconstruct High Speed and High Dynamic Range Videos from Events. CVPR 2021.

---

> ### Comment · Reviewer_CTBx · 2025-08-05
>
> The authors have addressed my concerns.

---

> ### Author Response · Authors · 2025-08-05
> **Thanks for your response**
>
> We sincerely thank you for your positive feedback and for confirming that your concerns have been addressed! We appreciate your constructive suggestions, which have helped us enhance the quality and clarity of our manuscript. We will ensure all promised revisions are incorporated into the final version if accepted.

---

### Comment · Area_Chair_ZHNF · 2025-08-04

Dear reviewers,

The authors' rebuttal has been uploaded. Could you please read the responses carefully and check if there are any questions or concerns that you would like to further discuss with the authors?

The author-reviewer discussion phase is going to end soon this week (Aug 6 11:59pm AoE).

Best,

Your AC

---

### Note · Authors · 2025-08-12

First of all, we would like to express our gratitude to the Area Chair and all the reviewers for your time and effort in reviewing our paper! Most of the reviewers recognized that "the proposed idea is very interesting and insightful" and "the introduction of new datasets is a valuable contribution".

We thank all the reviewers for their engagement in the discussions!
- We provide detailed discussions on the concerns raised by **Reviewer CTBx** including novelty of framework, practical limitations, and challenging scenarios. After reading our rebuttal, the reviewer confirmed that "the authors have **addressed my concerns**".
- **Reviewer hHyQ** raised concerns including experimental results and the necessity of events. We followed the suggestions to add more recent comparison methods (i.e., DAC and HybridDepth), quantitative results on 4DLFD-Semi-Real dataset, and an ablation study to demonstrate our arguments. The reviewer decided to "keep the **positive rating**" after the discussions.
- **Reviewer kC23** requested for quantitative results on 4DLFD-Semi-Real dataset and more recent comparison methods. We have added a recent baseline method (i.e., DAC) and all available results accordingly. However, the reviewer decided to "maintain the original score" due to concerns "regarding the allocation of space in the paper". We respectfully believe this is a **minor adjustment** and will reorganize the paper to include more details in final version of main text if accepted.
- The concerns of **Reviewer 1JE8** include novelty of framework, comparison fairness and information leak. As we clarify the method design and add an ablation study to exclude the influence of pretrained checkpoints, the reviewer acknowledged "the authors have addressed all my concerns", and decided to "**raise my score**".

All of the valuable suggestions provided by the reviewers will be incorporated to further enhance the quality of our paper. Specifically, we will:
- adjust the paper structure to include more details in the main text;
- add more discussions on the practical limitations, necessity of events and information leakage;
- include new comparison methods, ablation studies, and quantitative results discussed in our rebuttal;
- update the tables to indicate the type of compared methods;
- revise the texts to make them correct, concise and more clear.

---

### Decision · Program_Chairs · 2025-09-17

**Decision:**

Accept (poster)

**Comment:**

A paper introduces a metric depth estimation approach by fusing events with image foundation models via a prompting approach.
The paper received **three borderline accept** and **one borderline rejects**.

**Strengths**

- (CTBx,1JE8) The method is well-motivated and carefully designed
- (hHyQ,kC23,1JE8) Proposing a new dataset, good empirical results

**Weaknesses**
- (CTBx,hHyQ,kC23) Clarity on technical details (ablation study on hyperparameters, limitations, details of data construction)
   - (resolved) partially addressed in the rebuttal (reviewer kC23 disagrees)
- (CTBx) Unclear real-world applicability due to the hardware complexity of the event-RGB system
   - (resolved) Addressed in the rebuttal.
- (CTBx,1JE8) Not novel ideas (eg., prompting image foundation model with sparse depth.)
   - (resolved) Rewriting to differentiate the paper from related work
- (hHyQ) Evaluation on 4DLFD-Semi-Real and EDFV-Real datasets
   - (resolved) Rebuttal included the evaluation

All reviewers confirm that the issues or concerns were resolved in the rebuttal. Reviewer kC23 said the concerns are still remaining after the rebuttal. After carefully checking it, the concerns about additional quantitative results, comparison with recent methods, writing, etc, are resolved. It is highly recommended to accommodate all the discussions and analyses in the final version.

**AC recommends accepting the paper.**